# Ancient genomes from the Himalayas illuminate the genetic history of Tibetans and their Tibeto-Burman speaking neighbors

Chi-Chun Liu[1], David Witonsky[1], Anna Gosling [1,8], Ju Hyeon Lee[2], Harald Ringbauer [1,9], Richard Hagan[3,10], Nisha Patel[4,11], Raphaela Stahl [5], John Novembre [1], Mark Aldenderfer [6✉], Christina Warinner [5,7✉], Anna Di Rienzo [1✉] & Choongwon Jeong [2✉]

Present-day Tibetans have adapted both genetically and culturally to the high altitude environment of the Tibetan Plateau, but fundamental questions about their origins remain unanswered. Recent archaeological and genetic research suggests the presence of an early population on the Plateau within the past 40 thousand years, followed by the arrival of subsequent groups within the past 10 thousand years. Here, we obtain new genome-wide data for 33 ancient individuals from high elevation sites on the southern fringe of the Tibetan Plateau in Nepal, who we show are most closely related to present-day Tibetans. They derive most of their ancestry from groups related to Late Neolithic populations at the northeastern edge of the Tibetan Plateau but also harbor a minor genetic component from a distinct and deep Paleolithic Eurasian ancestry. In contrast to their Tibetan neighbors, present-day non-Tibetan Tibeto-Burman speakers living at mid-elevations along the southern and eastern margins of the Plateau form a genetic cline that reflects a distinct genetic history. Finally, a comparison between ancient and present-day highlanders confirms ongoing positive selection of high altitude adaptive alleles.

[1] Department of Human Genetics, University of Chicago, Chicago, IL 60637, USA. [2] School of Biological Sciences, Seoul National University, Seoul 08826, Republic of Korea. [3] Department of Anthropology, University of Oklahoma, Norman, OK 73019, USA. [4] Department of Plant and Microbiology, University of Oklahoma, Norman, OK 73019, USA. [5] Max Planck Institute for Evolutionary Anthropology, 04103 Leipzig, Germany. [6] Department of Anthropology and Heritage Studies, University of California, Merced, CA 95343, USA. [7] Department of Anthropology, Harvard University, Cambridge, MA 02138, USA. [8]Present address: Department of Anatomy, University of Otago, Dunedin 9054, New Zealand. [9]Present address: Department of Human Evolutionary Biology, Harvard University, Cambridge, MA 02138, USA. [10]Present address: Department of Archaeology, University of York, York YO10 5DD, UK. [11]Present address: Kintai Therapeutics, Cambridge, MA 02139, USA. ✉email: maldenderfer@ucmerced.edu; warinner@shh.mpg.de; dirienzo@bsd.uchicago.edu; cwjeong@snu.ac.kr

The Tibetan Plateau is characterized by hypobaric conditions, rough terrain, cold temperatures, and a relatively low biological productivity. Despite these constraints, ethnic Tibetans have successfully adapted to this environment and have lived on the plateau for millennia[1]. Understanding their genetic and cultural adaptations to this challenging hypoxic environment is of great archaeological, anthropological, genetic, and physiological interest[2]. To fully do so requires answering many fundamental questions regarding the origins of present-day Tibetan populations, including the source populations and initial movements of peoples onto the Plateau, the timing of the establishment of permanent Plateau populations, and the establishment of the gene pools ancestral to the present-day Tibetans.

Although archaeological data relating to early population movements onto the Plateau are sparse, the Baishiya Karst Cave site (3280 masl, meters above sea level) on the extreme northeastern edge of the Tibetan Plateau suggests the presence of Denisovan-related peoples between 160 and 60 thousand years ago (kya)[3–5]. Dates at the Nywa Devu site (4600 masl) on the central Plateau suggest a modern human presence between 30 and 40 kya[6]. Whether either of these sites reflects a permanent settlement of humans on the Plateau is unknown. Meyer et al.[7] propose an initial permanent occupation of the central Plateau at Chusang (4270 masl) by hunter-gatherers between 7.4 and 12.7 kya. In contrast, Chen et al.[8] and others have argued that a permanent population on the central Plateau was not possible until the advent of barley-based agriculture around 3.6 kya. The latter model generally presumes that agriculture was introduced onto the Plateau by migrants from lower elevation sites (<2500 masl) along the northeastern margins of the Plateau; these migrants are proposed to have contributed substantially to the gene pool of present-day Tibetans[9].

However, evidence for more complex, multiple origins of present-day Tibetans is also supported by genetic data. Densely sampled uniparental markers can be traced for the most part to lineages present in northern East Asia since the early Holocene, but older haplogroups such as mitochondrial M16 and Y chromosomal D-M174, originating from a deep Eurasian lineage, are also uniquely present among present-day Tibetans[10–12]. The idea of an ancient Paleolithic contribution to the Tibetan gene pool has also been proposed based on whole genome sequence data. A study comparing present-day Tibetan genomes to those of ancient Siberians and archaic hominins inferred a contribution from a mixture of ancient ancestries—archaic and non-archaic— among the hypothesized early peoples on the Plateau[13]. This proposal is consistent with the finding of a haplotype at the *EPAS1* (Endothelial PAS Domain Protein 1) locus that introgressed from a Denisovan-like population into the present-day Tibetan gene pool, conferring a selective advantage in high altitude environments[14–17].

Taken together, current genetic data suggest a multi-stage settlement of the Plateau: movements of Pleistocene-era populations with some level of archaic admixture onto the Plateau followed by Holocene-era migrations from the northeastern edges of the Plateau. Although the identity and origins of the Pleistocene-era population remain unknown, a recent analysis has identified a clear east-west cline of genetic variation within present-day geographically dispersed Tibetan populations[18]. This cline may reflect Neolithic population movements, such as those that might have been associated with the spread of barley agriculture. Prior to the spread into the Plateau, barley agriculture was practiced by Late Neolithic and Early Bronze Age populations in the Gansu-Qinghai region, such as those associated with the Qijia culture (ca. 2300–1800 BCE)[19,20]. This cline may also have been established or reinforced by later historical events, such as the expansion of the Tibetan empire since the 7th century CE, or by a prolonged process of gene flow between nearby populations in an isolation-by-distance manner that did not involve long-range migrations.

Ancient DNA (aDNA) data has the potential to resolve these questions, in part because genetic inferences from ancient populations are not confounded by recent historical events. Previous aDNA studies of individuals from three high elevation Himalayan sites in the Mustang district of north-central Nepal dating to 800 BCE–650 CE showed that these sites were inhabited by populations of clear East Asian ancestry who had likely migrated from the Tibetan Plateau[21].

Here we obtain aDNA data from additional individuals from these and four additional Himalayan sites in the Mustang and Manang districts (MMD), increasing the temporal coverage by more than 600 years, from ca. 1420 BCE–650 CE, and providing the earliest genetic evidence to date for Plateau populations. We show that these ancient Himalayan populations genetically cluster with present-day Tibetans and that they represent an early branch within the Tibetan lineage, making them particularly informative for inferring the history of the Tibetan gene pool, its origins, and its current distribution among the present-day Tibetans and their neighbors.

## Results

**Ancient genomes from the Himalayas.** Here we analyze genome-wide data of 38 ancient individuals from seven sites in the MMD region, Nepal (Fig. 1; Supplementary Data 1–3): Suila ($n = 1$; 1494–1317 BCE), Lubrak ($n = 2$; 1269–1123 BCE), Chokhopani ($n = 3$; 801–770 BCE), Rhirhi ($n = 4$; 805–767 BCE), Kyang ($n = 7$; 695–206 BCE), Mebrak ($n = 9$; 500 BCE–1 CE), and Samdzong ($n = 12$; 450–650 CE). Of these 38 individuals, 31 are newly reported in this study and seven were previously reported in a prior study of the region[21]. We also produced new data for two of the previously published individuals, resulting in new genome-wide data for 33 individuals (Supplementary Data 1, 2). All data were generated from human dental material. Due to disturbance of mortuary contexts, some teeth were initially assumed to be from distinct individuals but were later identified as replicate samples based on genetic data, resulting in nine individuals with data from multiple teeth (Supplementary Data 4). Data from multiple teeth and libraries belonging to a single individual were pooled accordingly prior to downstream analyses. Among the seven archaeological sites, Suila, Lubrak, Rhirhi, and Kyang have not been previously described (Supplementary Text 1). After initial genetic screening, 13/33 individuals were whole genome sequenced to low coverage (0.5-6.6x per individual; Supplementary Data 1). We additionally applied capture-enrichment methods to target two sets of single nucleotide polymorphisms (SNPs): (1) a set of "1240K" variants, designed to intersect with markers on the Affymetrix Human Origins and the Illumina genotyping arrays[22] and here captured for all 33 individuals; and (2) an additional set of 50 K variants, selected and curated from selection scan and phenotype association signals in present-day Tibetan populations[23] and captured for 21 individuals (Supplementary Data 1). The combined per-individual data satisfied standard quality control measures for ancient genomic data (Supplementary Data 2). For downstream analysis, we assembled two reference datasets derived primarily from published genome-wide genotype data produced on the Affymetrix Human Origins ("HO"; ~500 K SNPs) and the Illumina ("Illumina"; ~220 K SNPs) genotyping arrays (Supplementary Data 5, 6). We augmented these datasets with published ancient genomes as well as genomes of present-day Sherpa and Tibetan individuals from Nepal (Supplementary Data 5, 6). Whereas we focused most of our analyses on the HO set for its higher SNP density, we also used the Illumina set for in-depth analysis of diverse Himalayan populations across Nepal, Bhutan, India, and Tibet Autonomous Region[24].

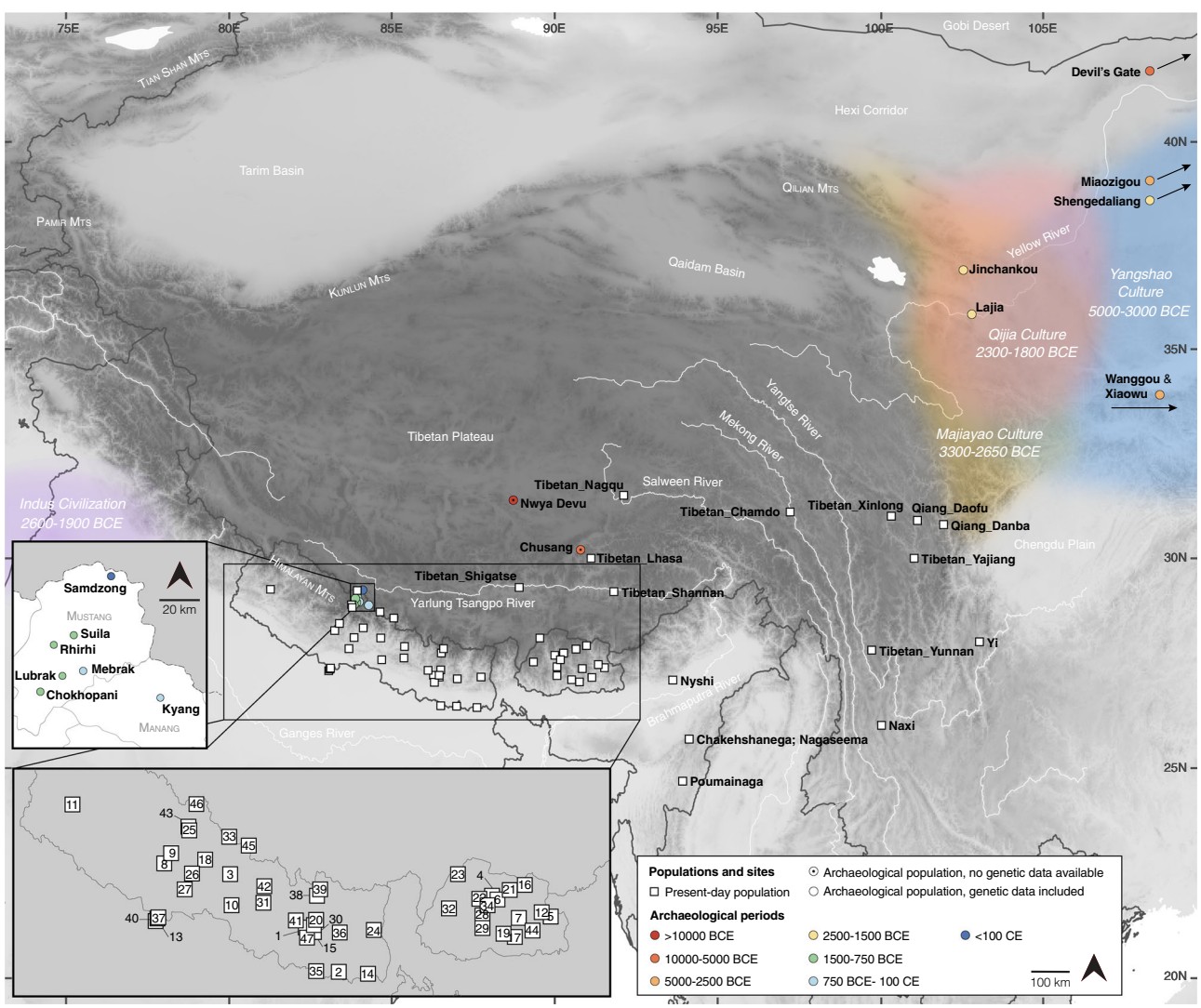

**Fig. 1 Geographic locations for ancient groups and present-day Tibeto-Burman speakers.** Circles represent ancient groups and are colored by archaeological periods; squares represent present-day populations of Tibeto-Burman speakers. Left inset: an enlarged view of the seven aMMD sites. Lower left inset: an enlarged view of the present-day Nepalese and Bhutanese populations: 1. Bahing; 2. Bantawa; 3. Baram; 4. Brokkat; 5. Brokpa; 6. Bumthang; 7. Chali; 8. Chamling; 9. Chantyal; 10. Chepang; 11. Chetri; 12. Dakpa; 13. Damai; 14. Dhimal; 15. Dumi; 16. Dzala; 17. Gongduk; 18. Gurung; 19. Khengpa; 20. Kulung; 21. Kurtop; 22. Lakha; 23. Layap; 24. Limbu; 25. Lower_Mustang; 26. Magar; 27. Majhi; 28. Mangde; 29. Monpa; 30. Nachiring; 31. Newar; 32. Ngalop; 33. Nubri; 34. Nup; 35. Puma; 36. Sampang; 37. Sarki; 38. Sherpa; 39. Sherpa_Khumbu; 40. Sonar; 41. Sunwar; 42. Tamang; 43. Thakali; 44. Tshangla; 45. Tsum; 46. Upper_Mustang; 47. Wambule. The base map was created in R v4.0.0 using publicly available map and altitude information from the mapdata v2.3.0 and elevatr v0.3.4 packages.

**The genetic structure of high altitude East Asians and their neighbors.** To describe the genetic profile of the ancient individuals from Nepal (aMMD) in the context of world-wide human diversity, we first performed principal component analysis (PCA)[25]. After confirming that they cluster with other East Asian individuals (Supplementary Fig. 1), we projected the aMMD individuals onto the first two PCs calculated for present-day Eastern Eurasian individuals (Fig. 2; Supplementary Data 4). The present-day populations form a structure with three spurs representing, respectively, clines of ancestry corresponding to southern Chinese and southeast Asians (SC-SEA), northeast Asians, and Tibeto-Burman populations. The Ami of Taiwan, Ulchi of the Lower Amur River basin in the Russian Far East, and Sherpa of Nepal form the distal ends of the three spurs, respectively. The Tibeto-Burman spur matches the east-west genetic cline of present-day Tibetans reported in a previous study[18]. Consistent with our previous results[21], all aMMD individuals, including those from the newly investigated sites of Suila, Lubrak,

Rhirhi and Kyang, cluster together with present-day Tibetan populations. The genetic profiles obtained from the unsupervised model-based clustering method ADMIXTURE are consistent with those from the PCA, with aMMD individuals sharing unique ancestral components with mid and high altitude present-day populations (Supplementary Fig. 2). Likewise, outgroup-$f_3$ statistics[26] indicate that the aMMD individuals have the highest level of shared genetic drift with each other, followed by present-day Sherpa and Tibetans, and then by low-altitude Tibeto-Burman speakers such as Naxi, Yi, and Nagaland populations in India (Supplementary Figs. 3, 4).

The uniparental haplogroups of the aMMD individuals also support their close genetic relationship with present-day Sherpa/Tibetans (Supplementary Data 2). We assigned Y haplogroups for 14 aMMD individuals. We observed little diversity, with 13 of the 14 males having derived markers of the Y-haplogroup O-M117, and 12 males carrying derived markers of its sublineage Oα1c1b-CTS5308 (Supplementary Fig. 5)[27]. Among present-day

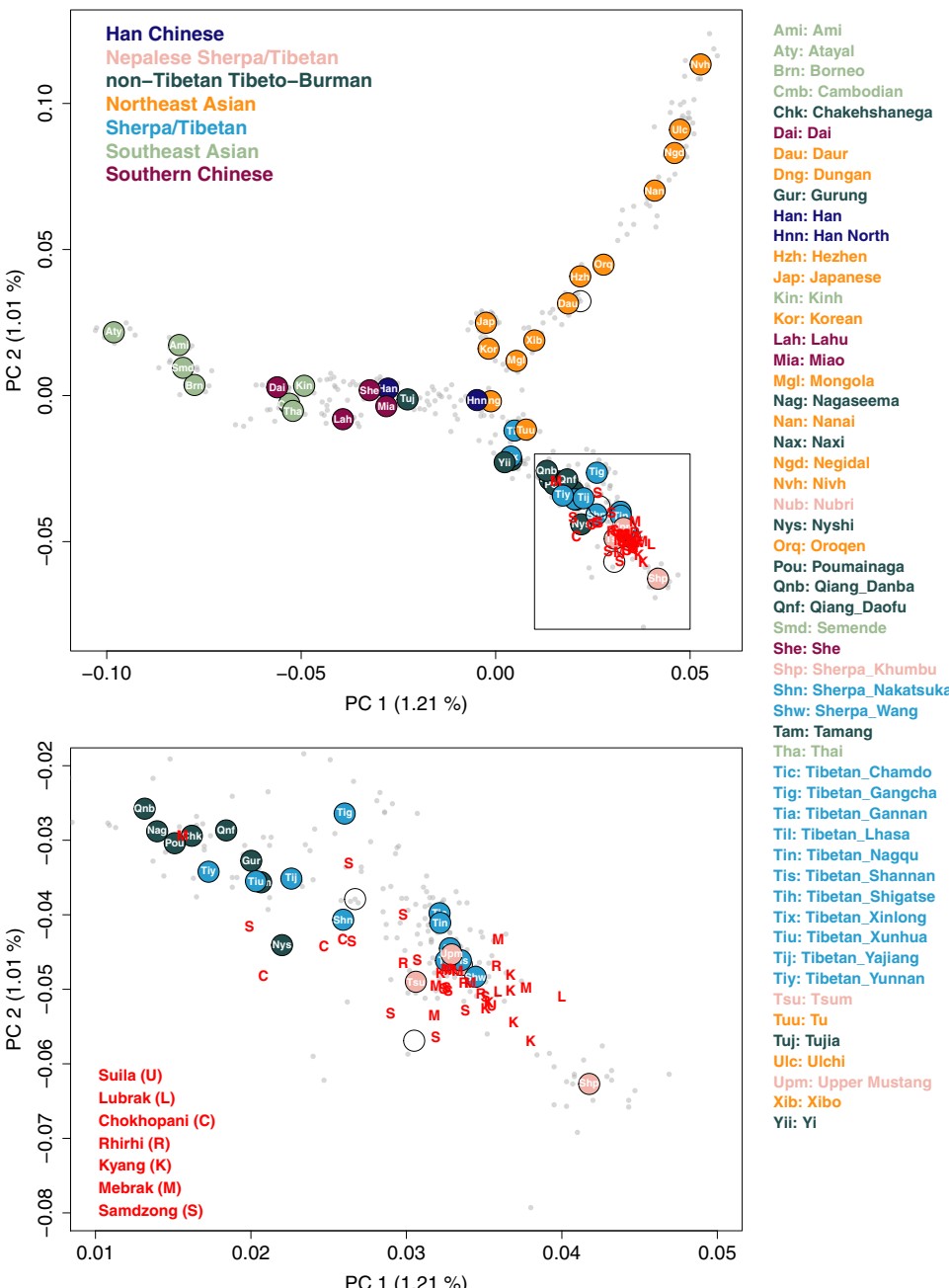

**Fig. 2 aMMD individuals on the top two PCs of present-day Asian individuals.** We calculated PCs from 486 present-day Asian individuals in the HO dataset and projected aMMD individuals on top of the top PCs. Gray dots represent present-day individuals we used to calculate PCs. Circles represent median positions of present-day groups colored by their language families along with their respective group abbreviations. Red capital letters "U, L, C, R, K, M, S" represent projected aMMD individuals.

populations, this sublineage is found primarily among Tibetans and Sherpa on the Plateau, in contrast to its sister lineage Oα1c1b-Z25929, which is today mainly found in Southern China and Northeast India[27]. A rapid radiation of all extant O-M117 lineages is estimated to have occurred 7000–5000 BP and has been interpreted as reflecting the spread of Sino-Tibetan languages, likely originating from northern China[27]. Notably, the Y-haplogroup O-M117 has been found also in ancient individuals from the upper Yellow River Neolithic Yangshao and Late Neolithic Qijia cultures[28], providing evidence for the majority of male aMMD lineages tracing back to this region. One aMMD male individual (S41) belonged to a different Y-haplogroup, D1a, which is another common haplogroup on the Tibetan Plateau today[10]. The

mitochondrial haplogroups of the aMMD individuals, while more diverse, are also prevalent among present-day Tibetans (Supplementary Data 2).

**The genetic relationship between ancient and present-day high altitude East Asians.** While most closely related to each other (Supplementary Figs. 3, 4), the aMMD individuals show subtle differences in their genetic affinity that may suggest a fine-scale genetic heterogeneity among them (Supplementary Fig. 3). Most prominently, all aMMD groups have the highest outgroup-$f_3$ statistic with Lubrak, while having the lowest value with Chokhopani. Indeed, all the other aMMD groups, including the earliest Suila, are

significantly closer to individuals at Lubrak than Chokhopani, as measured by $f_4$ (Mbuti, aMMD; Chokhopani, Lubrak) (>+4.4 SEM, standard error measure). The same pattern is also observed for present-day Nepalese Sherpa/Tibetans (>+2.7 SEM), while lowland East Asian populations are symmetrically related to Chokhopani and Lubrak (Supplementary Data 7). Using qpWave, we formally compared the two topologies ((Lubrak, aMMD), Chokhopani) and ((Chokopani, aMMD), Lubrak). We show that Suila, Rhirhi, Mebrak, and Samdzong are cladal to Lubrak (i.e., the former topology holds) within the limits of our resolution ($p > 0.192$), and Kyang only slightly differentiates from Lubrak ($p = 0.027$; Supplementary Table 1). In contrast, modeling the aMMD groups as a sister group of Chokhopani uniformly failed and thus the latter of the two topologies can be rejected ($p < 1.38 \times 10^{-4}$). A combination of Lubrak with a minor contribution from a South Asian group (e.g., Pulliyar) adequately fits all four groups, with an estimated South Asian ancestry contribution of only 1.9–5.1% ($p > 0.179$; Supplementary Table 2). For Chokhopani, neither Lubrak + South Asian nor Lubrak + Naxi/Yi/Naga fits ($p < 3.67 \times 10^{-4}$); however, Suila + Naxi/Yi/Naga fits with a substantial lowlander contribution (31–40%; Supplementary Table 3). We also detect a significant signal of admixture in Chokhopani using DATES, which infers an admixture time in Chokhopani of $46 \pm 11$ generations before the time of Chokhopani, placing it at ca. 1500-2800 BCE (for mean $\pm 2$ SEM; Supplementary Fig. 6). This implies gene flow must have occurred between Chokhopani and the ancestors of these low/middle altitude populations prior to 800 BCE, and plausibly before 1500 BCE.

Like aMMD groups, present-day Sherpa/Tibetan groups from the MMD region and the nearby Gorkha/Solukhumbu districts in Nepal[23], as well as Tibetans from more distant locations, are genetically closest to Lubrak and then to each other among the ancient and present-day East Asians (Supplementary Fig. 7; Supplementary Data 8). The earliest aMMD group from Suila, as well as later aMMD groups, are also among the top outgroup-$f_3$ signals of present-day Sherpa/Tibetans. Chokhopani shows smaller outgroup-$f_3$ values as expected from its admixture signal with lowlanders. Therefore, we conclude that Lubrak/Suila are so far the earliest known representative of a gene pool that is most enriched in the high altitude populations in the Tibetan Plateau and the Himalayas; we refer to this gene pool as the "Tibetan" lineage in this study.

**A dual genetic origin of high altitude East Asians.** Archaeological data suggest that Neolithic populations of the Upper/Middle Yellow River basin exerted a major cultural influence on the spread of farming onto the Plateau[8]. This region has also been proposed as the likely homeland of the Sino-Tibetan language family[29,30]. Interestingly, among ancient lowland East Asians[28,31–33], Middle/Late Neolithic groups from the Upper Yellow River region and its periphery (Fig. 1) show the closest genetic affinity to the aMMD groups (Supplementary Fig. 8). These include Late Neolithic individuals from the Jinchankou and Lajia sites in the Upper Yellow River region belonging to the Qijia culture (ca. 2300-1800 BCE; Upper_YR_LN), individuals from the Late Neolithic Shimao site of Shengedaliang in Shaanxi province (ca. 2250-1950 BCE; Shimao_LN), and those from the Middle Neolithic Miaozigou site in Inner Mongolia (ca. 3550-3050 BCE; Miaozigou_MN). These three groups have a similar genetic profile, deriving ~80% of their ancestry from a gene pool related to the Middle Neolithic individuals of the Yangshao culture sites of Wanggou and Xiaowu in the Central Plain (ca. 4000-3000 BCE; YR_MN) and the remaining ~20% from the Ancient Northeast Asian (ANA) gene pool related to Neolithic-era hunter-gatherers from the Devil's Gate Cave site of the Russian Far East ("DevilsCave_EN")[28,32]. Taking Upper_YR_LN and YR_MN as

representatives of lowland gene pools, we modeled the relationship between aMMD and Upper_YR_LN/YR_MN via a graph-based approach using qpGraph[34]. YR_MN fails to mimic the primary source of the aMMD groups and present-day Sherpa/Tibetans, mainly due to the extra affinity of aMMD to the ANA gene pool (Supplementary Table 4). In contrast, Upper_YR_LN, having a stronger genetic affinity to ANA, is consistently chosen as their primary genetic source in the best-scored graphs (Fig. 3). Together with their geographic and temporal proximity with early farmers on the Plateau, our results support a major genetic link between Plateau populations and the predecessors of early barley farmers on the northeastern fringe of the Plateau. However, we note that this genetic link was already established in the earliest aMMD groups dating to 1494–1317 cal. BCE at the far southern end of the Plateau (Supplementary Data 3). This date is only ~200 years after the proposed onset of the ca. 1650 BCE barley farmer expansion from the northeastern fringe of the Plateau[8]. A rapid population expansion from the Yellow River across the entire Plateau, a distance of more than 1800 km across rough terrain, would need to be invoked to explain these findings. Hence, substantial genetic exchange with lowlanders likely occurred prior to the barley expansion.

Although deriving 80–92% of their ancestry from a lineage related to Upper_YR_LN (Supplementary Table 4), the aMMD and present-day Sherpa/Tibetans are not adequately modeled as a sister clade to Upper_YR_LN, as expected given the unique genetic components of Tibetans not shared with lowlanders, including the *EPAS1* allele from a Denisovan-related admixture. Rather, the remaining 8–20% of their ancestry derives from a deep part of the population graph near the split between Western and Eastern Eurasian branches (Fig. 3; Supplementary Fig. 9). This source, however, does not derive from archaic hominins (Neanderthals or Denisovans, who contribute <0.5% genome-wide ancestry), and our results reject previously suggested sources of gene flow into the Tibetan lineage[13,35,36], including deeply branching Eastern Eurasian lineages, such as the 45,000-year-old Ust'-Ishim individual from southern Siberia, the 40,000-year-old Tianyuan individual from northern China, and Hoabinhian/Onge-related lineages in southeast Asia (Supplementary Fig. 10), suggesting instead that it represents yet another unsampled lineage within early Eurasian genetic diversity. This deep Eurasian lineage is likely to represent the Paleolithic genetic substratum of the Plateau populations.

**Two-routes of dispersal of the Tibeto-Burman speakers to the Himalayas.** The south-facing slopes of the Himalayas harbor many ethnolinguistic groups that show a striking pattern of stratification across altitudes: Indo-Iranian speaking South Asian populations occupy the lowlands, Sherpa/Tibetans occupy the highlands, and various non-Tibetan Tibeto-Burman speaking groups, such as the Tamang and Gurung, occupy the middle altitude range[37,38]. While Sherpa/Tibetans in Nepal likely arrived in the Himalayas from the Plateau (i.e., the northern route)[39], a previous genetic study suggested a separate southern route of migration for the middle altitude Tibeto-Burman groups[37]. However, how non-Tibetan Tibeto-Burman speaking groups are related to each other and to the Tibetan lineage has remained unclear. Here we utilize Lubrak, the most representative ancient group within the Tibetan lineage, to investigate the genetic history of Tibeto-Burman speaking populations. Specifically, we model Sherpa/Tibetans and other Tibeto-Burman groups using Nepalese Tibetans from Tsum as one source and Upper_YR_LN/YR_MN as the other, while using Lubrak as a key outgroup to distinguish the Tibetan lineage from lowlander ancestries with high resolution. Consistent with a previous report[18], we observe that the Tibetan groups from the Plateau and the Himalayas form a genetic cline. First, Nepalese Tibetans from the

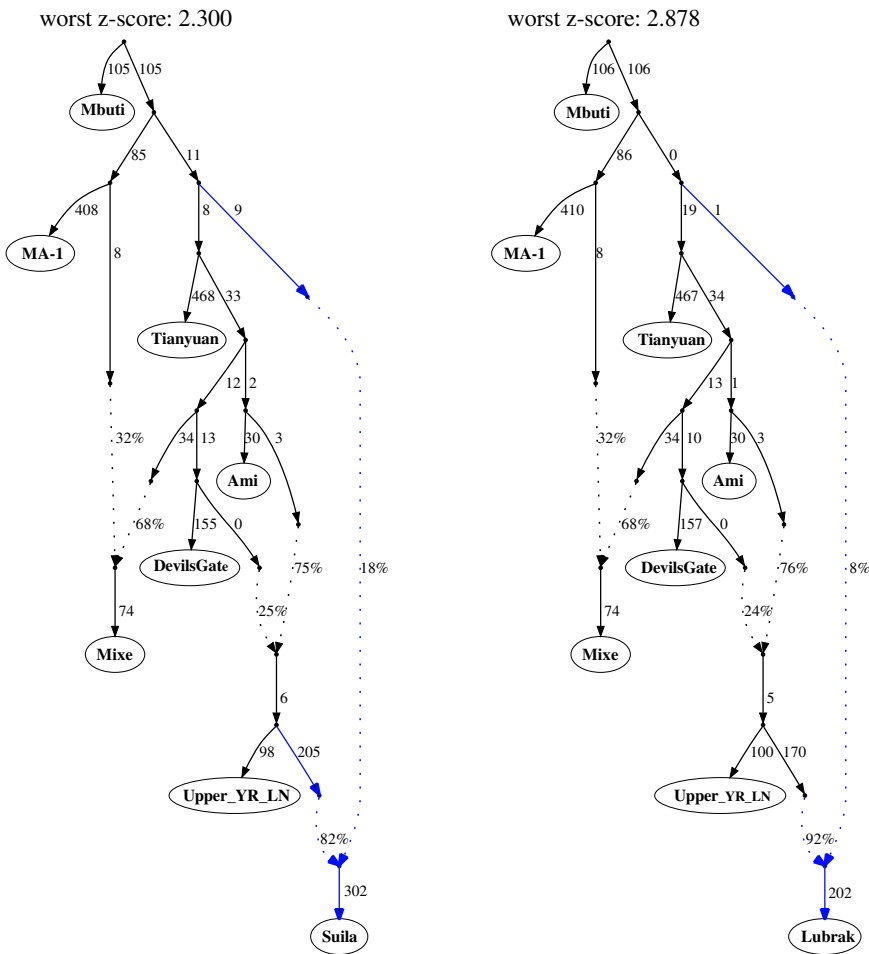

**Fig. 3 Admixture graph modeling for aMMD groups using qpGraph.** aMMD groups are modeled as two-way mixtures with Upper_YR_LN as one source and a deep lineage as the other source. The phylogenetic position of the deep lineage is inferred to be around the split between western and eastern Eurasian lineages but no further specification could be made due to limited resolution of our dataset. Here we present a graph for Suila that prefers a deep eastern Eurasian source and one for Lubrak that has zero-length branch suggesting affinity to neither western nor eastern Eurasian lineages. Alternative topologies and those without a deep Eurasian gene flow are presented in Supplementary Fig. 9. Z-scores are calculated by 5 cM block jackknifing as implemented in the qpGraph program.

Mustang and Gorkha districts (Upper Mustang, Nubri, Tsum), which are cladal to each other, and Sherpa from the Solukhumbu district derive 87–92% of their ancestry from the Tibetan lineage, which is represented by Tsum (Fig. 4; Supplementary Table 5). Second, Tibetans relatively close to the Himalayas (e.g., Lhasa, Shigatse, Shannan) derive a major proportion of their ancestry from the Tibetan lineage (76–86%). Last, Tibetan groups further to the east or northeast have much higher contributions from the low-lander lineage (21–58%). While we know from radiocarbon dating that the two poles of this cline, represented by aMMD and Upper_YR_LN, were already present by ca. 1420 BCE, the admixture process between the two poles that formed the present-day cline may have occurred later. Additional archaeogenetic studies in the Plateau are needed to understand when the cline began to form and how it developed over time across the Plateau.

With respect to the non-Tibetan Tibeto-Burman populations, we infer genetic links among them along the circum-Plateau route (Fig. 5). We describe the results beginning with the southeast edge of the Plateau with the Naxi and Yi and proceed clockwise to the southwest (Fig. 5). First, Naxi and Yi from southwestern China have a genetic profile that closely resembles that of YR_MN but distinct from that of Upper_YR_LN (Supplementary Fig. 11). Using qpAdm, we model Naxi/Yi as a sister clade of YR_MN with no contribution from the Tibetan lineage required (Supplementary

Table 5). Models using Upper_YR_LN as a proxy fail by returning ancestry coefficients larger than 1 from Upper_YR_MN. Naga from northeastern India are modeled as a mixture of 68–78% YR_MN/ Naxi/Yi and 22–32% Tibetan lineage (Fig. 5; Supplementary Tables 5–6). Finally, Tamang and Gurung from the mid-altitude region of the southern Himalayas have even higher levels of their ancestry from the Tibetan lineage (60–63%), as well as a South Asian influx (9–19%) in addition to the YR_MN-like ancestry (Supplementary Table 7). Models using Naxi/Yi/Naga as a source instead of YR_MN also fit (Supplementary Table 7). This same three-way admixture model, Tibetan lineage + YR_MN/Naga + South Asian, also adequately fits the 16 Bhutanese Himalayan groups previously published[24], with heterogeneous levels of contribution from the YR_MN/Naga (21–47% YR_MN or 32–75% Naga) and Tibetan lineages (20–82%; Supplementary Fig. 12; Supplementary Data 9). Overall, the South Asian contribution is small but non-negligible for many Bhutanese groups, ranging from 0 to 7%. Interestingly, for the populations in Nepal with substantial South Asian ancestry (e.g., Baram, Chantyal, Chepang, Gurung), south Indian tribal groups (e.g., Pulliyar) better represent their South Asian ancestry than northern Indian groups (Supplementary Data 9). These results highlight the complexity and multi-layered admixture history of Tibeto-Burman populations in the Himalayas.

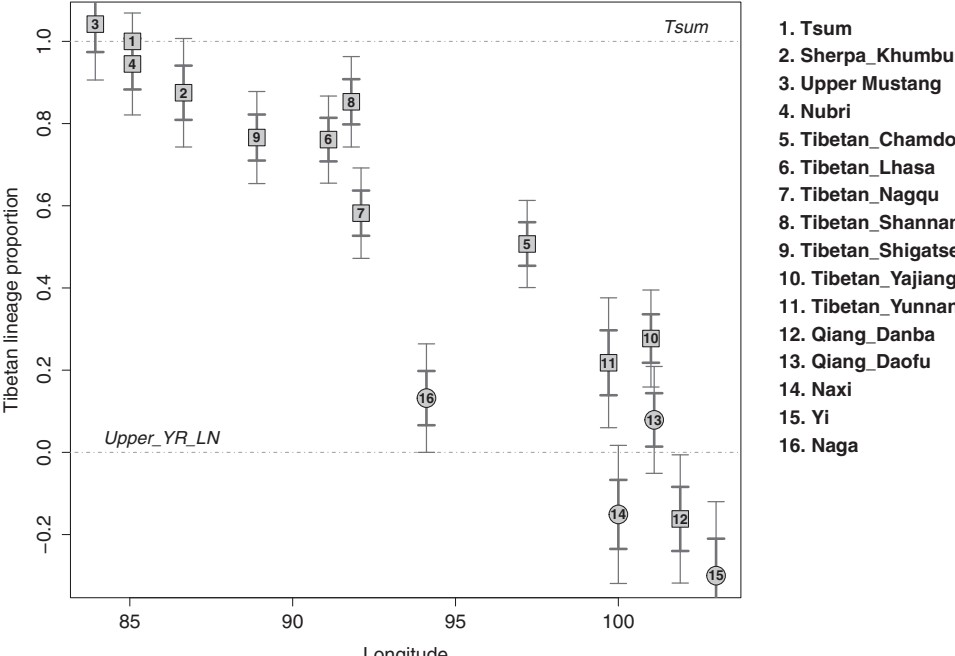

**Fig. 4 A genetic cline of Tibeto-Burman groups.** We model Tibeto-Burman groups using Nepalese Tibetan from the Tsum region ("Tsum") and Upper_YR_LN as the two sources using qpAdm. Tibetans from the plateau and Tibetans close to the Himalayas derived the majority of their ancestry from the Tibetan lineage, while Tibeto-Burman groups further to the east derived a much higher proportion of their ancestry from the lowlander lineage. The numbered circles/rectangles represent point estimates from qpAdm, and the thick and thin vertical segments represent ±1 and ±2 standard error measures (SEM) estimated by 5 cM block jackknifing, respectively.

**Prolonged positive selection on the *EPAS1* and *EGLN1* regions in Tibetans**. Our previous study reported that derived alleles for positively selected SNPs in the *EPAS1* gene were observed only in the later Samdzong individuals but not the older Chokhopani and Mebrak individuals[21]. Including our new aMMD genomes, we still do not detect derived alleles in the *EPAS1* haplotype block in the Chokhopani and Suila individuals, but we observe them at intermediate frequency in the other five sites (25–58%). Interestingly, the derived allele frequency in the ancient samples overall is lower than in present-day Tibetans (75%), indicating that selection still acted upon these alleles in the recent past (Supplementary Fig. 13; Supplementary Tables 8, 9). We also attempted to investigate the frequency changes at two adaptive nonsynonymous alleles in the *EGLN1* gene: rs12097901, which is common among East Asians, and rs186996510, which is virtually unique to Tibetans[16,40]. Unfortunately, unfavorable capture conditions limit coverage for these two SNPs. Nevertheless, reads from shotgun sequencing suggest that the frequency of derived alleles in the genomic window spanning the *EGLN1* gene in the aMMD samples is similar to that of present-day Tibetan populations (Supplementary Table 8); whether this finding indicates that selection on the *EGLN1* alleles did not extend over the time period covered by the aMMD samples or is simply due to the sparsity of the sequence data is unclear.

We next took advantage of 18 shotgun sequenced individuals in this study and our previous study[18] to perform a genome-wide selection scan with window-based $f_3$-statistics[41] (Fig. 6; Supplementary Table 10). The method quantifies allele frequency differences between ancient and present-day Tibetans, using Han Chinese as an outgroup, and therefore aims to detect positive selection in present-day Tibetans since the time of the aMMD specimens. Combining 17 ancient individuals (excluding one individual due to relatedness), the genomic windows overlapping the *EPAS1* gene show the strongest signals, supporting the continued positive selection at this locus. The genomic windows overlapping the *EGLN1* gene show the second

strongest signals. Interestingly, the elevated $f_3$-statistics in these windows are not driven by the nonsynonymous SNPs rs12097901 and rs186996510 that had already reached high frequency in aMMD, but instead by SNPs that are common in both aMMD and Han but are rare in present-day Tibetans (Supplementary Data 10). Next, we looked at the overlap between the signals found in this selection scan (using *z*-score threshold of 4) and a previous set of signals (top 0.1% PBS values genome-wide) identified by using only contemporary population data[23]. All but three of the overlapping signals appear to be contributed to by the strong signature at the *EPAS1* and *EGLN1* loci (Supplementary Data 11). Of the three remaining regions, two span the *PET112* and *MCL1* genes, which are not well-established candidates for the response to hypoxia, and one contains the *AKT3* gene, which is involved in angiogenesis and is implicated in the control of red blood cell traits in a candidate gene study[42].

## Discussion

In this study, we analyze the genetic profile of 38 ancient Himalayan individuals and show that the ancestry found today among high altitude East Asians (i.e., Tibetans and Sherpa) was already distinctly diverged from lowlanders by 1494–1317 BCE. This pushes back the earliest evidence for the Tibetan gene pool at least by 500 years from our previous reports on Chokhopani[21]. Leveraging these early genomes, we illuminate key features of the genetic history of Tibetans and their relatives in the Tibetan Plateau and its periphery. We find that the Tibetan lineage is well-modeled as a mixture of two genetic ancestry sources: one is an ancient and previously uncharacterized Paleolithic substratum which accounts for up to 20% of contemporary Tibetan ancestry, and the other is related to lowlanders living at the northeastern fringe of the Plateau during the Late Neolithic. The Paleolithic substratum appears to have contributed exclusively to the Tibetan gene pool among the present-day populations studied to date.

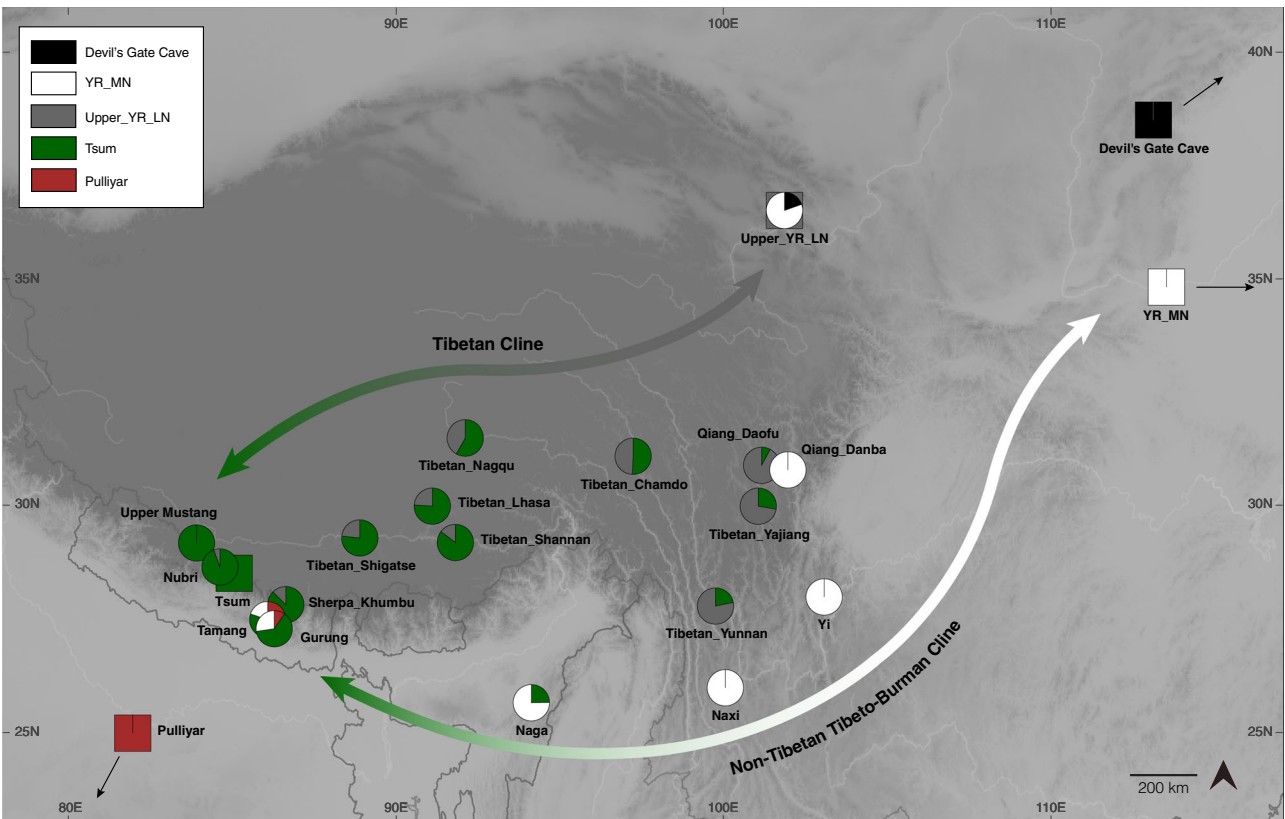

**Fig. 5 Genetic links between Tibeto-Burman speakers.** Tibetan groups from the Plateau and the Himalayas form a genetic cline, with the two poles represented by present-day Nepalese Tibetans (as well as aMMD) and Upper_YR_LN ("the Tibetan cline"). The non-Tibetan Tibeto-Burman cline reflects admixture along the circum-Plateau route and includes mid-altitude populations such as Naxi, Yi, Naga, Tamang and Gurung. Naxi and Yi cannot be modeled as a part of the Tibetan cline, i.e., Tsum+Upper_YR_LN; instead, YR_MN alone adequately models them. Non-Tibetan Tibeto-Burman speakers have higher contribution from the Tibetan lineage (represented by Nepalese Tibetan Tsum), and far-western mid-altitude populations Tamang and Gurung further have South Asian influx. Squares indicate the source populations used in ancestry models (circles).

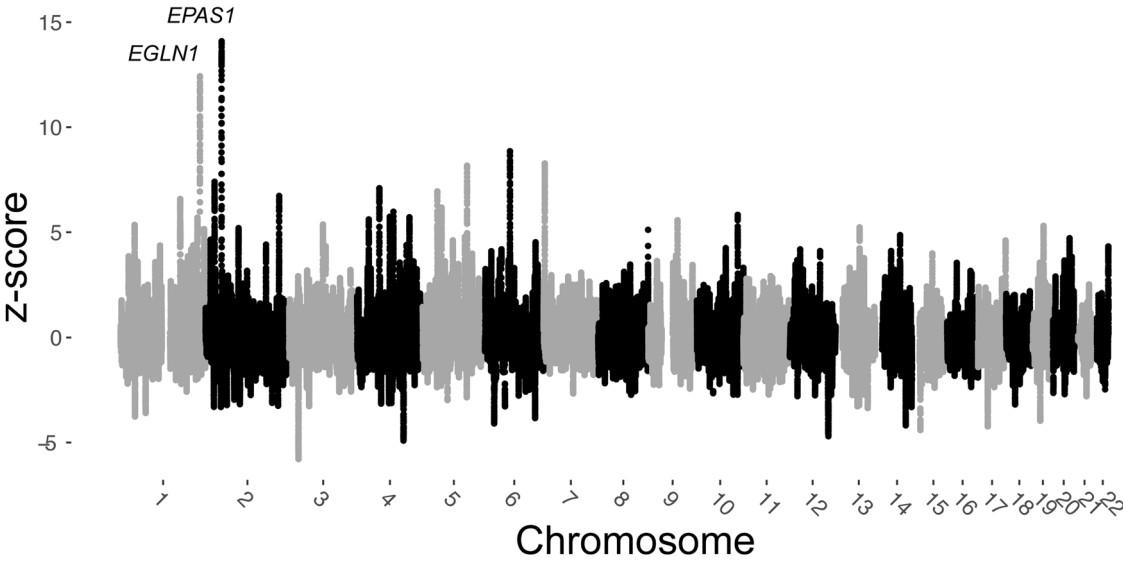

**Fig. 6 Genome-wide selection scan using outgroup-$f_3$ statistics with sliding windows.** We computed $f_3$ (Tibetans; aMMD, Han) using a sliding window approach with a window size 500 kb and a step size 10 kb. $Z$-scores for each window were calculated with a resampling approach (see Methods). Windows spanning the *EPAS1* and *EGLN1* genes harbor the two top signals.

Our extensive modeling of present-day Tibetan and non-Tibetan Tibeto-Burman speakers identifies two genetic clines to explain their genetic history. These clines presumably reflect two distinct routes of population dispersal that are reflected in the distribution of diverse Tibeto-Burman languages in the Himalayas: one traversing the Plateau from its northeastern fringe to the Himalayas (the "northern" route), and the other along the periphery of the Plateau and the southern fringe of the Himalayas

(the "southern" route) (Fig. 5). We provide a formal admixture modeling of the Tibetan populations along the northern cline and corroborate our previous report of this cline[18,43]. The genetic, cultural and linguistic diversity of present-day Tibeto-Burman speakers along the southern slope of the Himalayas reflects the confluence of ancient populations arriving via these two routes following their separation since the Late Neolithic.

The unique features of the Tibetan genetic profile have long puzzled researchers, leading to wildly different and often incompatible population history models, ranging from Tibetans representing a sister clade that split from Han Chinese less than 3000 years ago[14], to Tibetans branching off from a Han Chinese-related lineage more than 9000 years ago with gene flow from Paleolithic Siberians (Ust'-Ishim) or even from an unknown archaic hominin[13,35]. Moreover, these previous models, contradicting to each other, were developed on the basis only of present-day Tibetans and Han Chinese data and accepted an overly simplistic assumption that both populations are representative of the ancient groups ancestral to the two major branches of the Sino-Tibetan language family, i.e., Tibeto-Burman and Sinitic, respectively. Here we utilized ancient genomes from key time periods and geographic locations, which are better representatives of the lineages being modeled than present-day populations, to perform a direct test of the proposed demographic models.

In our study, we show that ancestors of present-day Tibetans have been present in the Himalayas since at least ca. 1420 BCE, when the earliest direct evidence for sustained human presence appears at aMMD sites such as Suila and Lubrak. Moreover, we confirm the close relationship between early Himalayan populations and Late Neolithic groups living along the northeastern fringe of the Plateau around 2300–1800 BCE (Upper_YR_LN). Neolithic groups in the Gansu-Qinghai region likely include the ancestral population of those who later expanded onto the Plateau; however, the precise timing of the expansion is not clear. Barley cultivation, which is more suitable to the cooler and drier climate of the Plateau than millet, has long been argued to have allowed the Neolithic expansion onto the Plateau. While our results may ostensibly fit the long-held hypothesis of barley-driven expansion onto the Plateau ca. 1650 BCE, such a massive demic diffusion from Qinghai to the Himalayas in only about 200 years is unlikely to be a sole explanation for the ancient genetic link between the Plateau and the Gansu-Qinghai region. We propose an alternative scenario in which the genetic link between Plateau and lowland populations may have formed much earlier and therefore may not have been related to the introduction of barley or other domesticated plants or animals of Western Eurasian origin. The Karou site in eastern Tibet (ca. 5000–3000 BP) and the Qugong site near Lhasa (ca. 3800–3000 BP) show an indigenous archaeological tradition, and have assemblage composition and ceramic motifs distinct from those at Qijia[2]. In addition, the evidence from the Zongri site (ca. 2600–2000 BCE) suggests that Plateau hunter-gatherers traded for millets with lowlanders much earlier than the presumed introduction of barley[44]. The absence of an *EGLN1* selection signature in the Upper_YR_LN combined with an estimated *EGLN1* selective sweep dated to around 8000 BP[40,45] suggests that the two populations may have already split long before the arrival of barley in Gansu-Qinghai. Barley was cultivated as a minor crop in Gansu-Qinghai as early as ca. 2000 BCE, leaving open the possibility for an earlier barley-driven expansion prior 1650 BCE, but archaeological evidence to support such a scenario is lacking. We acknowledge that our present data cannot completely reject the barley hypothesis; therefore, we call for a search of ancient genomes from the Plateau older than 1650 BCE to directly test it.

Finally, our study shows the prolonged effects of natural selection in shaping the gene pool of high altitude East Asians. Of

note, the increase in the *EPAS1* allele frequency over the time period spanning the aMMD samples and present-day Tibetans highlights the slow but steady action of positive selection on this Denisovan-derived genetic variant. Future studies on additional ancient genomes across the Tibetan Plateau will be able to lead us toward a comprehensive understanding of the evolutionary history of the two Tibetan signature genes, *EGLN1* and *EPAS1*, as well as to investigate further the polygenic signatures of adaptation suggested by the study of present-day genomes[23].

## Methods

**Ethics statement**. All specimens reported in this manuscript were exported to co-author M.A. under the authority of the Department of Archaeology (DoA), Government of Nepal via the permits to the Sky Door Foundation Nepal/USA from 2007-to the present. Sampling of archaeological samples in the field was overseen by on-site representatives of the DoA at Rhirhi, Lubrak, Suila, Kyang, and Samdzong. Representatives of Village Development Committees and other members of descendant communities were present during field sampling of archaeological materials at Rhirhi, Lubrak, Suila, and Samdzong. Those present were shown the material being taken for export and no objections were offered. Sampling of material from the Kapilvastu Museum was overseen by local staff and a representative of the DoA. Sampling of materials of the Mebrak remains was overseen at the DoA in Kathmandu. Approval for destructive analysis and genetic investigation was provided by the DoA as part of the permitting and export process. Outreach to descendant communities included: (1) local curation and control of excavated materials at the villages of Samdzong, Chuksang (for Rhirhi), Lubrak (materials housed and available in the Bon temple in the village), and Suila (materials housed in the Buddhist monastery at Ghiling) under the guardianship of community leaders; (2) small museums have been created at Samdzong and Chuksang for the display of materials excavated from Samdzong and Rhirhi; (3) a coloring book about archaeological science methods and featuring the site of Samdzong was developed by C.W. et al. and has been translated into Nepali and Tibetan by Ghiling community members Nawang Tsering Gurung and Tsering Dorjee Gurung and made available to local communities. These can be found online at https://doi.org/10.17617/2.3367799 (Nepali) and https://doi.org/10.17617/2.3367804 (Tibetan) and are distributed free of charge; and (4) all required reports have been submitted to the DoA; the Sky Door Foundation/Nepal is preparing to issue short summaries (in Nepali, possibly in Tibetan) of major project findings which will be offered to secondary schools in Jomsom and Lo Manthang as well as larger villages in Upper Mustang.

**Excavation of the aMMD archaeological sites**. The original depositional context at Chokhopani was destroyed by the installation of a micro-hydro pipeline. Data recovery was limited to the collection of artifacts and human remains that were discovered in the cave complex or were found downslope from it[46]. The Suila site was discovered in 2018 after the cave was damaged by road construction. Local villagers collected the exposed materials and took them to a monastic complex in Ghiling. These materials were shown to our survey crew shortly after the destruction of the site; they were photographed and samples were taken under the supervision of the representative of the DoA. No excavation was possible. The archaeological materials from the Lubrak site were discovered in 2018 following the erosion of a riverbank in the eponymous village. Villagers rescued the material which was taken to a monastery. Our survey team mapped and photographed the tombs from which the materials were recovered and documented the contents of the tombs. Samples of human remains were taken under the supervision of the representative of the DoA. The tombs remain in situ but no further data recovery has been undertaken. The Rhirhi site was discovered during archaeological survey in 2016. The site had been looted in the past and some cultural material was found upon entry into the cave. The profile of the looter's pit was cleaned, drawn, and additional samples of cultural material were discovered in the profile during that process. Mebrak 63 was thoroughly excavated and documented by a Nepali-German team[47]. Preservation of organic materials, including human remains, was excellent. Much of the interior of the site was covered in a 40-cm deep layer of bird guano which was carefully removed. Many of the remains in the tomb were commingled by repeated visits to add the recently deceased to the tomb. A total of eight palimpsest levels were mapped and then carefully excavated by hand; the matrix was not screened. The context was extensively documented with line drawings and photographs. The Kyang site, looted in the past, was first documented photographically. A 1 m grid was placed over the deposit and surface materials, including human bone, wooden, and other organic materials, were bagged by grid unit. Excavation proceeded using arbitrary levels of 10 cm; two levels were recorded and the deposit was excavated to bedrock. The soil matrix was screened using a fine mesh. The large majority of artifacts and other remains were recovered from the surface; few were found in either of the arbitrary levels[48]. The Samdzong site[48,49] consists of 10 caves, most likely shaft tombs, excavated into the sheer face of a cliff. Seismic activity had collapsed the tombs and the original context of deposition was churned and mixed with soil and rock from above. Cave interiors were generally quite shallow; each was documented photographically and

materials visible on the surface were collected and bagged. Due to the mixed nature of the deposit, excavation was not done in stratigraphic levels. All excavation was done by hand, and larger artifacts encountered were bagged as discovered. The matrix of the caves was screened with fine mesh; this enhanced the discovery of smaller artifacts such as scraps of metal, fragments of animal and human bone, wood, and numerous glass beads.

**Absolute dating of the archaeological sites and samples**. Accelerator Mass Spectrometry (AMS) [14]C dating of 15 samples are newly reported in this study (Supplementary Data 3): Suila ($n = 1$), Lubrak ($n = 1$), Rhirhi ($n = 2$), Mebrak ($n = 2$), Kyang ($n = 2$), and Samdzong ($n = 7$). For Suila, Lubrak, Rhirhi, human teeth used in the genomic study were directly dated at the University of California, Irvine W.M. Keck Carbon Cycle AMS facility (lab code UCIAMS). For Mebrak, Kyang, Samdzong, uncarbonized wood or animal tooth specimen were dated at the Curt-Engelhorn-Zentrum Archäometries-Zentrum (lab code MAMS). All [14]C dates were calibrated using the Northern Hemisphere calibration curve in the online version of Calib 8.2 (http://calib.org/calib/calib.html) (Supplementary Data 3).

**Sample selection**. A total of 54 suitable teeth and one petrous bone (from U2) were selected for analysis from seven archaeological sites in the MMD region: Suila ($n = 2$; 1494–1317 BCE), Lubrak ($n = 2$; 1269–1123 BCE), Chokhopani ($n = 3$; 801–770 BCE), Rhirhi ($n = 6$; 805–767 BCE), Kyang ($n = 11$; 695–206 BCE), Mebrak ($n = 10$; 500 BCE–CE 1), and Samdzong ($n = 20$; CE 450–650) (Supplementary Data 1–3). The samples were selected to augment previously published whole genome data from eight individuals at Chokhopani ($n = 1$), Mekbrak ($n = 3$), and Samdzong ($n = 4$)[21]. However, bioturbation and faunal disturbance of the human remains in some cases made it challenging to unambiguously assign skeletal and dental material to discrete individuals. Because of this, data from all 62 aMMD skeletal materials were reanalyzed, of which 55 were sufficiently well-preserved to perform genetic relatedness testing, resulting in the identification of 42 distinct individuals, of which 34 are new to this study (Supplementary Fig. 14; Supplementary Data 4). Three teeth analyzed in this study were determined to originate from two individuals (M63 and S10) who were previously published[21].

**DNA extraction**. Ancient DNA (aDNA) handling for all samples was performed in dedicated aDNA facilities at the University of Oklahoma (OU) and the Max Planck Institute for the Science of Human History (MPI-SHH). Both laboratories operate HEPA-filtered aDNA cleanrooms (rated ISO7 or better) and DNA manipulation was additionally performed within dedicated laminar flow hoods. Cleanroom laboratory access and workflows are restricted, and personal protective equipment (consisting of full body Tyvek suits, gloves, masks, and goggles/face shields) is worn by all personnel. All surfaces are routinely sterilized with dilute bleach (NaOCl) solution and UV irradiated daily, and all PCR and post-PCR activities are performed in separate laboratory facilities. DNA extraction for the new samples from Chokhopani, Rhirhi, Kyang, Mebrak, and Samdzong ($n = 50$) was performed at OU. Briefly, tooth surfaces were wiped with 2% NaOCl, followed by mechanical abrasion and UV irradiation to remove surface contaminants. Tooth dentine (100 mg) was crushed to a coarse powder and digested in a 1 mL solution of 0.45 M EDTA and 0.25 mg/mL proteinase K. DNA was purified and concentrated using a Qiagen MinElute/Zymo reservoir apparatus following a published protocol[50]. In brief, 13 mL of PB buffer was combined with DNA supernatant in the reservoir apparatus and centrifuged to bind the DNA to the silica column membrane. The column was washed twice with PE buffer, followed by a dry spin. Purified DNA was eluted from the column with 60 µL of EB buffer.

DNA extraction for samples from Suila and Lubrak ($n = 4$) was performed at the MPI-SHH using similar methods with slight modifications. Sampling followed a previously described protocol[51]. Briefly, teeth were mechanically cleaned and UV irradiated to remove surface contamination. Teeth were sectioned and dentine powder was obtained from the inner pulp cavity using a mechanical drill. DNA extraction was performed as previously described[52]. Briefly, dentine powder (50 mg) was digested in a 1 mL solution of 0.45 M EDTA and 0.25 mg/mL proteinase K. DNA was purified and concentrated using Roche High Pure Viral Nucleic Acid kit. In brief, 10 mL of binding buffer (5 M GuHCl, 40% isopropanol) and 400 µL 3 M sodium acetate was combined with DNA supernatant in the High Pure Extender Assembly and centrifuged to bind the DNA to the silica column membrane. The column dry spun, washed twice with wash buffer (20 mM NaCl, 2 mM Tris-HCL in ethanol), and then dry spun again. Purified DNA was eluted from the column in 100 µL of TET buffer (10 mM Tris, 0.1 mM EDTA, 0.05% Tween 20).

**DNA library construction and sequencing**. To screen for preservation, Chokhopani, Rhirhi, Kyang, Mebrak, and Samdzong sample DNA extracts were constructed into double-stranded, double-indexed Illumina libraries using a blunt-end protocol at OU using a NEBNext DNA Library Prep Master kit, as previously described[21]. In brief, DNA was end repaired using NEB end repair mix (containing T4 polymerase and T4 PNK), followed by Qiagen MinElute purification. Illumina-compatible P5 (IS1+IS3) and P7 (IS2+IS3) adapter mix[53] was added using Quick T4 ligase, followed by Qiagen MinElute purification. An adapter fill-in reaction was

performed using Bst DNA polymerase, followed by heat inactivation. Following a qPCR to determine library concentration, library completion was performed by PCR amplification in triplicate using P5 and P7 indexing primers[53,54] and KAPA HiFi+Uracil HotStart enzyme, followed by Qiagen QiaQuick DNA purification. A total of 49/50 extracts were successfully built into libraries, and then sequenced at the University of Chicago on an Illumina HiSeq 4000 using 2 × 75 bp chemistry to screen for preservation.

To screen the samples from Suila and Lubrak (which became available later in the project), we built DNA extracts into either double-stranded (for Lubrak) or single-stranded (for Suila), double-indexed Illumina libraries at the MPI-SHH following similar protocols but with slight modifications, as described in published protocols[55–57]. In brief, for double-stranded libraries, DNA was end repaired using end repair mix (containing T4 polymerase and T4 PNK), followed by Qiagen MinElute purification. Illumina-compatible P5 (IS1+IS3) and P7 (IS2+IS3) adapter mix[58] was added using Quick T4 ligase, followed by Qiagen MinElute purification. An adapter fill-in reaction was performed using Bst DNA polymerase, followed by heat inactivation. Following a qPCR to determine library concentration, library completion[56] was performed by PCR amplification using P5 and P7 indexing primers and Pfu Turbo Cx HotStart DNA polymerase, followed by Qiagen MinElute DNA purification. Indexed libraries were then reamplified using Herculase II Fusion enzyme[59], and then purified and pooled for sequencing using a Qiagen MinElute.

For single-stranded libraries, DNA was dephosphorylated and heat denatured, after which the first adapter (CL78/TL137) was ligated to the DNA using T4 ligase. Dynabeads MyOne Streptavidin C1 magnetic beads were bound to the libraries, washed with BWT-SDS solution (0.1 M NaCl, 0.01 M Tris-HCl, 0.001 M EDTA, 0.05% Tween 20, 0.5% SDS), and incubated in stringency wash buffer. The libraries were washed with BWT solution (0.1 M NaCl, 0.01 M Tris-HCl, 0.001 M EDTA, 0.05% Tween 20) and incubated with extension primer CL130, followed by Klenow fragment. The libraries were washed with BWT-SDS solution and incubated in stringency wash buffer (0.015 M NaCl, 1.5 mM trisodium citrate, 0.1% SDS). The libraries were washed with BWT solution and then incubated with T4 ligase to ligate the second adapter (CL53/CL73). The libraries were washed with BWT-SDS solution, and incubated in stringency wash buffer. The libraries were washed with BWT solution and incubated in TT buffer (10 mM Tris, 0.05% Tween 20) at 95 °C to release the DNA from the beads. After determination of library concentration using qPCR, the library was indexed[56] by PCR amplification using P5 and P7 indexing primers and Pfu Turbo Cx HotStart DNA polymerase, followed by Qiagen MinElute DNA purification. Indexed libraries were then reamplified[59] using Herculase II Fusion enzyme, and then purified and pooled for sequencing using a Qiagen MinElute. A total of 4/4 extracts were successfully built into libraries and then sequenced at the MPI-SHH on an Illumina HiSeq 4000 using 1 × 75 bp chemistry.

Taken as a whole, 47 of the new 54 samples yielded sufficient endogenous DNA (>0.1%) on screening for further analysis. Of these, 25 samples (21 individuals) were selected for a custom in-solution capture using oligonucleotide probes matching 50 K manually selected target sites with functional significance ('50K', see assay design description below). To ensure sufficient genome-wide coverage of ancestry informative markers for ancestry analysis across the entire sample set, we performed an in-solution capture for ~1.2 million informative nuclear SNPs ('1240K')[22,60] on all 47 well-preserved samples. However, to improve the library complexity of the 43 samples initially processed at OU, we first generated new double-stranded, double-indexed libraries for these samples at the MPI-SHH using the method described for the Lubrak samples. We applied the 1240K capture to all 47 samples, and sequenced them on an Illumina HiSeq 4000 using 1 × 75 chemistry and Illumina NextSeq 500 using 2 × 75 bp chemistry until we achieved sufficient coverage on the captured SNPs or depleted library complexity (Supplementary Data 1). After removing three samples that failed our quality control criteria (C2 for low coverage, M3490 and U2 for 4% and 9% mitochondrial contamination, respectively), 44/47 samples (33 individuals) were included in our analysis. Finally, 15 samples (13 individuals) were selected for whole genome deep sequencing (WGS). Seven of these samples were sequenced at Macrogen, Inc. using an Illumina HiSeq X10 with 2 × 75 bp chemistry, and 9 (including one sample overlapping with the U of Chicago samples) were sequenced at the MPI-SHH using an Illumina HiSeq4000 with 1 × 75 bp chemistry (Supplementary Data 1); WGS samples sequenced at the MPI-SHH were subjected to UDG-half treatment[61]. These data were then combined with 7 previously published deeply sequenced aMMD genomes (individuals C1, M63, M344, S10, S35, S40, and S41; excluding M240 for its outlier position in PCA) for subsequent analysis, resulting in a total of 20 individuals with whole genomes sequenced to a depth of 0.1-6.6x. In total, genome-wide ('1240K') ancestry data was generated for 33 individuals, functional SNP ('50K') data was generated for 21 individuals, and whole genome data was analyzed for a total of 20 individuals, resulting in 38 individuals in the final dataset (which includes individuals from this study and the previous study[21]).

**Sequence capture**. Enrichment for the 1240K panel SNPs was performed at the MPI-SHH according to previously described protocols[22]. A quantity of 1-2 µg was used in each capture. In brief, DNA libraries were diluted to approximately 200-400 ng/µL and mixed with blocking oligos (human Cot01 DNA, salmon sperm DNA, P5, P7). The library pool was denatured at 95 °C for 5 min, followed by 37 °C for 10 min, and then added to a 96-well plate pre-prepared with hybridization

buffer and biotinylated DNA probes. Hybridization occurred at 65 °C for 24 hours, after which the biotinylated probes were immobilized on Dynabeads MyOne Streptavidin T1 beads. Unbound DNA was removed by three high temperature washes using HWT buffer, followed by incubation in melt solution to release the DNA libraries. The DNA library supernatant was transferred to a new plate and bound to SeraMag Speedbeads. The bead solution was repeatedly washed with ethanol and dried. The beads were resuspended in TT buffer to release DNA, and after pelleting beads, the supernatant containing the libraries was collected. After qPCR quantification of the enriched libraries, the libraries were PCR amplified using Herculase II Fusion. The thermal profile used was: initial denaturation at 95 °C for 2 minutes, 30 cycles of 95 °C for 30 seconds, 60 °C for 30 seconds and 72 °C for 30 second, and a final extension for 5 minutes at 72 °C. The post-capture libraries were purified using SPRI beads and quantified using a NanoDrop (ThermoFisher). An aliquot of 100 ng post-capture library was reamplified using a reconditioning assay to remove heteroduplexes. The thermal profile used was: 1 cycle of 95 °C for 2 min, 58 °C for 2 min, and 72 °C for 5 min. Aliquots of the resulting libraries were pooled in equimolar amounts and purified on a Qiagen MinElute. Library pool concentrations were measured with a NanoDrop and TapeStation (Agilent) prior to sequencing.

Enrichment for the functional SNPs ('50K') was performed at U of Chicago using custom biotinylated RNA probes generated by MYBaits (Arbor Biosciences). The capture was performed as per the manufacturer's recommendations. A quantity of 200 ng of DNA was used in each capture. Hybridization occurred at 55 °C for 48 hours, after which the biotinylated probes were immobilized on Dynabeads MyOne Streptavidin C1 beads. The manufacturer's instructions were followed to wash away any unbound DNA, prior to PCR amplification using Kapa HiFi HotStart. The bead-bound DNA was used directly. The thermal profile used was: initial denaturation at 98 °C for 2 minutes, 14 cycles of 98 °C for 20 seconds, 60 °C for 30 seconds and 72 °C for 30 second, and a final extension for 5 minutes at 72 °C. The post-capture libraries were then purified via MinElute (Qiagen), eluting into 20 ul of EB buffer following the manufacturer's instructions. The fragment length distribution of the enriched libraries was investigated using the BioAnalyzer (Agilent) and the concentration checked using Qubit (Invitrogen) to allow pooling of the samples in equimolar amounts for sequencing. The samples were sequenced in batches of 6 on a lane of an Illumina HiSeq 4000 instrument using 2×100 bp chemistry.

**Sequence data processing (including QC).** Illumina adaptor sequences were trimmed off using AdapterRemoval v2.2.0[62], and PCR duplicates were removed using DeDup v0.12.2[63]. Reads were mapped to the human reference genome with decoy sequences (hs37d5) using BWA v0.7.12[64]. For all samples, we estimated mitochondrial contamination using the Schumutzi package. For whole-genome sequenced and/or 1240K-captured male individuals, we also estimated contamination utilizing haploidy on the X-chromosome using the ANGSD contamination module[65]. No whole-genome sequenced nor 1240K libraries have estimated mitochondrial or nuclear (males only) contamination higher than 5% (Supplementary Data 1).

**Uniparental haplogroup.** To determine the Y haplotype branch of male ancient individuals, we analyzed SNPs on the Y chromosome. For reference, we used markers from https://isogg.org/tree (Version: 13.238, 2018). We additionally merged in refined Y haplogroup O SNPs defined by Wang et al. 2018[27]. We screened for ancestral and derived reads of this set of SNPs in each of the ancient males within our dataset. Haplotype calls were based on manual inspection of ancestral and derived read counts per haplogroup branch, factoring in coverage and error estimates. As the conventions for naming of haplogroups are subject to change, we annotate haplogroups in terms of carrying the derived state at a defining SNP. To determine the mitochondrial haplogroup, we first determined the consensus sequence using the log2fasta script in the Schmutzi package with a quality threshold 10. We then assigned haplogroups to each of the consensus sequences using Haplogrep 2[66].

**Genotype calling and relatedness.** To mitigate the impact of post-mortem damages in genotyping, we trimmed 5 bps from both ends of the reads using the bam module of BamUtil v1.0.14[67]. For the 1240K SNPs, we created a pileup for each library using samtools[68] mpileup v1.9 with "-R" and "-B" flags. Then, to call "pseudo-haploid" genotypes, we randomly drew a single high-quality base (Phred-scaled base quality score 30 or higher) from a high-quality read (Phred-scaled mapping quality score 30 or higher) per library, using the pileupCaller program in the sequenceTools v1.4.0.5 (https://github.com/stschiff/sequenceTools). For the Suila individuals, we ran pileupCaller with the "--singleStrandMode" flag, which took a random sample for C/T SNPs from negative strand reads only and for G/A SNPs from positive strand reads only. Then, we detected libraries from the same individuals by calculating the pairwise mismatch rate (PMR) of pseudo-haploid genotypes for all pairs of libraries (Supplementary Fig. 14A): duplicate pairs show PMR values ~0.12 while pairs from unrelated individuals have ~0.24 (Supplementary Data 4), twice the value of the duplicates[69]. After detecting duplicates, we merged BAM files per individual and repeated the genotyping procedure to create per-individual pseudo-haploid genotype data. We then re-calculated PMR using

the per-individual data, identifying 7 first-degree and 5 second-degree relatives (Supplementary Fig. 14B; Supplementary Data 4). We further calculated genotype likelihoods from per-individual masked BAM files using the "SNPbam2vcf.py" script in the lcMLkin v0.5.0 program[70] and estimated IBD1 and IBD2 coefficients using the lcMLkin program, resulting in distinguishing the first-degree relatives into four parent-offspring pairs and three full siblings.

**Data compilation.** We compiled two reference datasets for population genetics analysis: the HumanOrigins ("HO") and the "Illumina" datasets. For the HO dataset, we merged genome-wide genotype data of present-day world-wide populations from the Affymetrix Axiom Genome-Wide Human Origins 1 array, including a large number of South and East Asians[23,36,71,72]. We then augmented this panel with whole genome sequenced present-day Tibetan and Sherpa individuals[23,43,73] and two high-coverage ancient individuals: a Mesolithic European hunter-gatherer from Luxemburg ("Loschbour")[71] and a 45,000-year-old individual from western Siberia ("Ust'-Ishim")[74]. We also included pseudo-haploid genotype data of the following ancient individuals: Natufian[75], Ganj Dareh Neolithic[76], Villabruna[77], Anatolia Neolithic[22], MA-1[26], Botai[78], Tyumen_HG and Sosonivoy_HG[76], previously published ancient East Asian genomes including Hoabinhian hunter-gatherers, Devil's Gate Cave, YR_MN, Upper_YR_LN[28,31–33], and the aMMD (Supplementary Data 5). For the Illumina dataset, we merged South Asian[79], Tibetan and Sherpa individuals in Nepal[23,43,73], Himalayan individuals[24] with individuals in the Human Genome Diversity Project[80] (Supplementary Data 6). We then merged Burmese and Thai individuals from the Simon Genome Diversity Project[73]. We included the same set of ancient individuals as in the HO dataset. We removed strand-ambiguous SNPs from the datasets.

**Principal component analysis.** For the HO dataset, we calculated PCs for two sets of present-day populations: (i) 2096 Eurasian individuals and (ii) 486 East Asian individuals (Fig. 2; Supplementary Fig. 1; Supplementary Data 5). For PCA, we used the smartpca v16000 program in the EIGENSOFT package v7.2.1[25], with the "lsqproject: YES" option for both sets, and additionally with the "shrinkmode: YES" option for the East Asian set. Individuals not included in calculating PCs are projected onto the PC space using these options. These two options account for shrinkage due to missingness and projection, respectively.

**ADMIXTURE.** We analyzed admixture profiles for the 486 East Asian individuals and the aMMD individuals using the ADMIXTURE[81] program v1.3.0. We removed SNPs with minor allele frequency lower than 1% and kept independent SNPs by linkage disequilibrium pruning SNPs using the "indep-pairwise 200 25 0.2" command in PLINK v1.9[82], leaving 374,924 SNPs in the analysis. For each value of K from two to six, we performed ten runs with random seeds. We visualized the inferred ancestry components (the Q matrices) using pong[83].

**F-statistics.** We used the qp3Pop v650 and qpDstat v970 programs in the admixtools package v7.0[25] for computing outgroup-$f_3$ and $f_4$ statistics, respectively. For the HO dataset, we computed $f_3$ (Mbuti; X, Y), where X is an aMMD group and Y's include 54 present-day East Asian populations, 7 aMMD groups, and 28 published ancient East Asian groups[28,32,33] (Supplementary Fig. 3; Supplementary Data 5). Using the same set of Y's, we also conducted $f_4$ symmetry tests in the form of $f_4$ (Mbuti, Y; Chokhopani, Lubrak), $f_4$ (Mbuti, Y; Chokhopani/Lubrak, present-day Nepalese Sherpa/Tibetan), $f_4$ (Mbuti, Y; YR_MN/Upper_YR_LN, Naxi/Yi), and $f_4$ (Mbuti, Devil's Gate; YR_MN/Upper_YR_LN, aMMD) (Supplementary Fig. 11; Supplementary Data 7, 8). We also calculated the same outgroup-$f_3$ statistics using the Illumina dataset with Y's including the Himalayan populations and East Asian populations. Standard errors are calculated by 5 cM block jackknifing as implemented in the qp3Pop and qpDstat programs.

**qpWave/qpAdm modeling.** We used the qpWave v1200 and qpAdm v1201 programs in the admixtools package v7.0 for testing cladality between two groups and for testing various two-way and three-way admixture models, respectively. In all tests using the HO dataset, we use a base set of six outgroups to distinguish major branches of Eurasian ancestries in high resolution: central African Mbuti ($n = 10$), Onge from the Andaman Islands ($n = 11$), central American Mixe ($n = 10$), Iran_N from the Neolithic Ganj Dareh site in Iran ($n = 8$), Upper Paleolithic European hunter-gatherer Villabruna ($n = 1$), and Taiwanese Ami ($n = 10$) (Supplementary Data 5). On top of this, we added an extra outgroup that belongs to the Tibetan lineage to increase statistical power to distinguish the Tibetan lineage from other Eastern Eurasian lineages: Suila, Lubrak, Chokhopani. QpAdm results for present-day populations are primarily based on the base set +Lubrak. For the Illumina dataset, we replaced Mixe with south American Karitiana ($n = 13$). To maximize data usage, we used a non-default option "allsnps: YES", which uses all available SNPs for calculating individual f-statistic rather than taking a common set of SNPs covered by target, sources, and outgroup populations. For the admixture modeling of present-day Tibetan and other Tibeto-Burman populations, we used Tsum[23] as the Tibetan-related reference, YR_MN/Upper_YR_LN/Naga/Naxi/Yi as lowland East Asian-related reference, and Pathan/Mala/Pulliyar as the South Asian-related sources (Supplementary Tables 1–3, 5–7). For the Illumina dataset, we replaced Mala with Sindhi (Supplementary Data 9).

**Admixture graph modeling**. We inferred the most plausible admixture graph topology with qpGraph v7365 in the admixtools package v7.0. We first built a five-population skeleton graph without admixture events from Mbuti, MA-1, Tianyuan, Ami, and DevilsGate. We then added Mixe and Upper_YR_LN, and an aMMD group or present-day $f_3$ Tibetan/Sherpa onto the topology sequentially. When adding a group, we enumerated all possible admixture graph topologies up to two-way admixture by choosing up to two existent branches as proximal source groups. We then chose the topology that led to the least number of $z$-scores > 2, where the scores are calculated by 5 cM block jackknifing. We also prefer topologies with positive internal branch lengths. Note that the procedure above is greedy in that a different order from which these populations were added can lead to a different final topology. We repeated our graph searching procedure by replacing Tianyuan with two Hoabinhian hunter-gatherers in southeast Asia ("McColl_SEA_GR1"; La368 and Ma911; Supplementary Data 5). We then tried adding archaic hominin (Altai Neanderthal and Denisovan) or Ust-Ishim into our five-population skeleton. Lastly, we tried adding a group merged from 2 Eneolithic Botai, Tyumen_HG and Sosonivoy_HG, hoping this group could tease out whether the deep lineage shows more East or West Eurasian affinity.

**Admixture dating**. We tested a gene flow from non-Tibetan Tibeto-Burman populations to Chokhopani and to estimate the date of this admixture using DATES v753[76], with options "jackknife: YES" and "binsize 0.01". We used Suila and a combined set of present-day Naxi and Yi as two sources.

**Selection scan**. We computed outgroup-$f_3$ statistics of the form $f_3$ (Tibetans; aMMD, Han) for selection scan. Allele counts for present-day Tibetans and Han were computed from genotypes of 27 whole-genome sequenced unrelated present-day Tibetan Nepalese and 103 Han (CHB) individuals from the 1000 Genome Project. Allele counts for aMMD were computed from 17 shotgun sequenced aMMD pseudo-haploid genomes (excluding KS8 due to relatedness from the 18 individuals). We applied a sliding window approach with a window size 500 kb and a step size 10 kb[41]. $F_3$ statistics in a window was computed from biallelic SNPs in the window that are segregating in either CHB or present-day Tibetans, provided there are more than 15 aMMD individuals having more than one read covering a SNP. We further excluded windows with <250 SNPs. These raw $f_3$ values were normalized by the heterozygosity of present-day Tibetans in the corresponding windows to mitigate their dependency on allele frequencies[34]. We randomly sampled one $f_3$ in each LD block[84] to estimate means and standard errors of the $f_3$ values and convert window-based $f_3$ values into $z$-scores.

**Reporting summary**. Further information on research design is available in the Nature Research Reporting Summary linked to this article.

## Data availability

The raw DNA sequences (FASTQ) and the alignment data (BAM) reported in this paper have been deposited in the European Nucleotide Archive (ENA) under the accession number PRJEB41752. The genotype data for the 1240K and HumanOrigins panels have been deposited in the Edmond Data Repository of the Max Planck Society [https://edmond.mpdl.mpg.de/imeji/collection/H8mcNv5pbSen6sLg?q=]. The 15 new AMS dates reported in this study, their associated lab codes, and their corresponding lab protocols are provided in Supplementary Data 3. Previously published genome-wide data of ancient individuals used in this study are listed in Supplementary Data 5 and are available via the following sources: (1) the combined genotype data for the 1240K panel SNPs provided by the Reich Lab [https://reich.hms.harvard.edu/datasets], (2) BAM files of the Devil's Gate Cave individuals in the ENA under the accession number PRJEB29700, and (3) the genotype data for the 1240 K panel SNPs of ancient individuals from China in the Genome Sequence Archive in the Beijing Institute of Genomics Data Center under the accession number HRA000123.

## Code availability

All of the analyses performed in this study are based on the publicly available softwares. Specific version information as well as non-default arguments are described in the Methods section.

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

## Acknowledgements

We would like to thank Rowan K. Flad for helpful comments on the earlier versions of the paper. We are also grateful to Karma Ngodup for helpful discussions during the course of this project. This work was supported by the National Institutes of Health (R01HL119577 to A.D.), the US National Science Foundation (BCS-1528698 to M.A.), National Research Foundation (NRF) of Korea grant funded by the Korea government (MSIT) (2020R1C1C1003879 to C.J.), the European Research Council under the European Union's Horizon 2020 research and innovation programme (grant agreement numbers 804884-DAIRYCULTURES to C.W.), and the Max Planck Society.

## Author contributions

C.J., A.D., C.W., and M.A. conceived and supervised the study. A.G., R.H., N.P., R.S. performed the laboratory works. M.A. and C.W. provided archaeological materials and associated information. C-C.L., C.J., D.W., A.G., J.L., H.R. analyzed data. C-C.L., C.J., A.D., C.W., M.A., J.N. wrote the paper with the input from all coauthors.

## Competing interests

The authors declare no competing interests.
