## [Peer Review File · Nature Communications]

Ancient genomes from the Himalayas illuminate the genetic history of Tibetans and their Tibeto-Burman speaking neighborsReviewers' Comments:

Reviewer #1:

Remarks to the Author:

In this manuscript, Liu et al. report new genome-wide data from 31 ancient individuals from the Himalayan region. Overall, this study greatly increases the ancient genomes availability for the region (previously 7 individuals) and the archaeological periods covered. This work plays an important contribution in better understanding the complex demographic history of the Himalayan region and it would be of general interest to readers of different fields.

I have a couple of general comments:

- 1) I would improve and expand Figure 5 and its caption to better summarise the demographic inferences reported in the manuscript. The results described here involve many populations and samples and it would help expert and non-expert readers to understand the general picture.
- 2) Many of the demographic modelling in the manuscript has been inferred by qpAdm, qpWave and qpGraph. I am not an expert on this, but I am wondering how confident the authors are regarding the population modelling and admixture proportions they report. From my understanding, the method requires the user to specify lists of target, source and reference populations. How did the author choose the populations to include in the model? What are the parameters used in these runs? This information should be more specifically stated in the manuscript.
- 3) In the selection scan what is the z-score that is considered significant? If $z\text{-score} > 4$ is considered significant, there are multiple signals in other chromosomes:
 - a. Have any of those regions already been reported in previous selection scans in modern/ancient samples? Did the authors consider to test for polygenic adaptation?
 - b. What is the frequency of those SNPs in modern Tibetans compared to ancient samples? How did the authors interpret values with negative z-score?
 - c. Did the authors consider to use other selection methods (for example XP-EHH)?

Other minor comments:

Page 3, lines 110-111: what's the difference between the newly generated data and the previous ones for the two previously published individuals?

Page 5, lines 164-165: an ADMIXTURE plot with both modern and ancient samples might be good to visualise fine-scale heterogeneity in the ancient samples and their relationships with modern individuals.

Page 5, lines 178-181: can this admixture event be dated using methods like DATES presented in Narasimhan et al, 2019 (<https://zenodo.org/record/3263997#.XRnebJNKj6A>) or ALDER?

Page 8, lines 299-301 and Table S17: What are the SNPs tested tagging the derived haplotype in EPAS1?

Figure 1: is there a way to indicate if samples are from high, low and middle altitudes?

Figures S2 and S3: I would label the populations with the highest sharing with the ancient samples on the plot or mention it in the caption.

Figure S4: I would insert a small legend in the figure explaining the different parts of the plot or expand the description in the figure caption.

Reviewer #2:

Remarks to the Author:

In " Ancient genomes from the Himalayas illuminate the genetic history of Tibetans and their Tibeto-Burman speaking neighbors" the authors generate genome wide sequencing and genotyping data on 33 ancient individuals (from approx. 3500 to 2000 years in the past) from high altitude regions on the southern end of the Tibetan plateau. Based on this genetic data they conclude that the population is

most closely related to Late Neolithic populations in the region, but also have a Paleolithic Eurasian ancestry. Late Neolithic origin is consistent with prior hypotheses that permanent settlements were only established on the northern edge of the Plateau due to the advent of barley-based agriculture. Paleolithic components of present day Tibetans is also seen in mitochondrial and Y chromosomal haplogroups. They also show by comparing ancient and modern samples, that positive selection has been acting throughout this time period on high altitude adaptive alleles.

The paper is well written and gives a thorough investigation of genetic similarities and source populations for ancient Tibetan and Tibeto-Burman samples. However, the methods and some of their assertions require further explanation/justification.

My comments are as follows:

The ancient data provides a snapshot into the past, but it does not make it a perfect sample of the past since it depends on from where and from what time periods the samples are sourced.

In the discussion, the authors state "However, the complex genetic histories of contemporary populations severely hamper the accuracy of inferences based on present-day population." However, from my reading, I fail to see how what has been proposed using present-day sample is inaccurate and how the proposed modeling done here provides more accurate inferences? It would be helpful if this could be done in a more statistically principled way.

The authors refer to incompatible inferences:

"Tibetans representing a sister clade that split from Han Chinese less than three thousand years ago" and "Tibetans branching off from Paleolithic Siberians (Ust'-Ishim) or even from an unknown archaic hominin." Could the authors say more as to how these are incompatible? The admixture graphs (from qpGraph) provide evolutionary relationships but not the timing of population splits. So, how does the graph reject a population split of 3000 years ago?

The authors show us a set of admixture graphs in the supplementary material (Figure S7 and 8), and it would be helpful to let the reader know specifically how these graphs reject previous incompatible results/hypotheses? There is also a set of assumptions and populations that are used to build these graphs that are not justified in the methods section. Also, it was not clear how the graph in the main text was chosen over all the other graphs in the supplemental section, given that these differ in the number of populations used, and model selection is a hard problem when the numbers of parameters differ. The authors claim that some graphs are a better fit than others but the comparisons are not systematically presented.

In the results section (lines 223-229), the authors state that "our results reject previously suggested sources of gene flow into the Tibetan lineage, including the deeply branching Eastern Eurasian lineages....in south east Asia (Fig. S8)." There are several graphs in Figure S8, and when I inspect the ones with archaic humans or Ust' Ishim, I see that there is no contribution from Denisovans into Tibetans, but we know that there is a Denisovan contribution. Is it because this method fails to detect low levels of admixture? I believe what the authors are referring to is results in Lu et al. (2019) - could the authors explain/justify more how these models reject those results?

How about some of the authors' previous work (e.g. Jeong et al. Admixture facilitates genetic adaptations to high altitude in Tibet. Nat. Commun. 5, 1-7 (2014))? How do their previous results stand in light of the new ancient DNA and new analyses presented here?

Other comments:

1) It would be good to have a clearer explanation/discussion of how the qpGraph modeling source populations were chosen for present day Tibetan admixture analysis (section starting Line 192 and Fig 3) - it appears to be related to pairwise genetic affinity estimates but it's not clear how such a complex model arose.

- 2) Regarding the unresolved deep Eurasian lineage that shows admixture in the aMMD groups - is there evidence for this admixed lineage by applying other tools to detect such features?
- 4) For clinal analysis of Figure 4, is Tsum being used also as a source (as noted in the text). If so that should be noted in the caption. It might also be more visually coherent/appealing to present this in the view of Figure 1 (with contours for the cline and source populations indicated)
- 5) how is f3 different than the population branch statistic?
- 6) Line 284: "Interestingly, the derived allele frequency in the ancient samples overall is lower than in present-day Tibetans (75%)." The frequency of Tibetans varies geographically, and I believe is correlated with altitude. What is the altitude of the samples being used for the present-day Tibetan populations?

Typos:

Line 233 - Two-routes *of* dispersal....

Reviewer #3:

None

Point-by-Point Response to Editor’s and Reviewers’ comments for NCOMMS-21-01104-T
“Ancient genomes from the Himalayas illuminate the genetic history of Tibetans
and their Tibeto-Burman speaking neighbors”

We received 24 comments from two reviewers. We present our response to each comment below.

REVIEWER COMMENTS

Reviewer #1 (Remarks to the Author):

Comment 1: In this manuscript, Liu et al. report new genome-wide data from 31 ancient individuals from the Himalayan region. Overall, this study greatly increases the ancient genomes availability for the region (previously 7 individuals) and the archaeological periods covered. This work plays an important contribution in better understanding the complex demographic history of the Himalayan region and it would be of general interest to readers of different fields.

We appreciate the reviewer’s positive assessment of our work.

I have a couple of general comments:

Comment 2: 1) I would improve and expand Figure 5 and its caption to better summarise the demographic inferences reported in the manuscript. The results described here involve many populations and samples and it would help expert and non-expert readers to understand the general picture.

We provide a new version of Figure 5 and a detailed legend to highlight our key findings described in the main text.

Comment 3: 2) Many of the demographic modelling in the manuscript has been inferred by qpAdm, qpWave and qpGraph. I am not an expert on this, but I am wondering how confident the authors are regarding the population modelling and admixture proportions they report. From my understanding, the method requires the user to specify lists of target, source and reference populations. How did the author choose the populations to include in the model? What are the parameters used in these runs? This information should be more specifically stated in the manuscript.

We describe our rationale for choosing the sources and the references, as well as the parameters we used, in the revised Methods section (“qpWave/qpAdm modeling”). We used “allsnps: YES” option to maximize data usage as the only non-default option for the qpWave/qpAdm analysis. In general, we modeled target groups as a mixture of the following three ancestry components (and their subsets): high-altitude East Asians, lowland East Asians, and South Asians. For lowland East Asian sources, we chose Upper_YR_LN, YR_MN, Naxi, Yi, and Naga for their geographic proximity and archaeological connection to the Tibetan plateau. For South Asian sources, we chose Pathan, Mala (or Sindhi for the Illumina data set), and Pulliyar to cover the north-south cline of South Asians. For high-altitude East Asian sources, we used Lubrak for the ancient aMMD targets, and Tsum (ethnic Tibetans from the Tsum region of the Gorkha district) for the present-day targets. We chose Lubrak because it is among the oldest aMMD group and it consistently shows the highest outgroup- f_3 statistics for all

aMMD groups. We chose Tsum for the present-day groups because it shows the highest affinity to the aMMD groups among the present-day populations and thus can be considered as a baseline.

We used the following base set of reference populations as described in the Methods section: Mbuti, Onge, Mixe, Iran_N, Villabruna, Ami. Mbuti is a non-Eurasian outgroup. Onge, Mixe, Ami are to distinguish between Eastern Eurasian ancestry components, mainly northeast Asian vs southeast Asian. Iran_N and Villabruna provide reasonable resolution to distinguish between various Western Eurasian ancestry components. Although choosing references is essentially a heuristic decision, we have used this set of references for several studies of ancient Eurasian populations and found it to provide high resolution. On top of this base set, we added one high-altitude East Asian group to maximize our resolution to distinguish high-altitude East Asian ancestry from lowland ones. For the present-day targets, we used Lubrak. For the aMMD groups, we used Lubrak, Suila, or Chokhopani when available.

Comment 4: 3) In the selection scan what is the z-score that is considered significant?

If z-score > 4 is considered significant, there are multiple signals in other chromosomes:

- a. Have any of those regions already been reported in previous selection scans in modern/ancient samples? Did the authors consider to test for polygenic adaptation?
- b. What is the frequency of those SNPs in modern Tibetans compared to ancient samples? How did the authors interpret values with negative z-score?
- c. Did the authors consider to use other selection methods (for example XP-EHH)?

The outgroup- f_3 statistic we used for the selection scan takes the form of $(p_{\text{Tibetan}} - p_{\text{aMMD}}) \times (p_{\text{Tibetan}} - p_{\text{Han}})$. Our main goal is to identify regions with recent positive selection: i.e. present-day Tibetans have a substantial allele frequency change compared to aMMD in the same direction as the Tibetan-Han difference.

- a. Formally designating a significance threshold for these tests is challenging; however, due to the low coverage of the ancient samples, we wanted to be very stringent, and rather than focus on a more typical threshold such as z-score >4, we chose to focus on the top 10 signals, corresponding to a z-score >7.5, which far exceeds the thresholds typically used. Nonetheless, we have now looked at the overlap between signals defined using a looser threshold (z-score >4) and a previous PBS scan using modern DNA data. We report the results in a revised text on p. 8. This analysis also addresses the question about SNP allele frequencies in modern Tibetans (reviewer comment 4b, first part).
Regarding polygenic adaptations, we had conducted a polygenic test using variants from traits that were reported in Jeong et al 2018 to have polygenic adaptation signals (Table S8) but did not find a significant result. We chose not to report this finding because it is not possible to determine whether it is the consequence of low power due to small sample size or a true negative.
- b. Regarding negative z-scores, we did not interpret these as evolutionarily interesting. Some fraction of negative z-scores are expected, simply because some variants will have less shared drift than the mean (recall z-scores are calculated after normalizing by the mean and standard deviation).
- c. Regarding methods such as XP-EHH and other linkage disequilibrium-based methods, we did not apply them to our low-coverage ancient genome data because they require diploid genotype data and minimally several dozen individuals for accurate calculation.

Other minor comments:

Comment 5: Page 3, lines 110-111: what's the difference between the newly generated data and the previous ones for the two previously published individuals?

The difference is that the new samples come from different teeth than the previous ones, and after an initial analysis, we found them to be genetically identical to previously published individuals from the site. Hence, we merged the data from these samples to increase the coverage per individual.

Comment 6: Page 5, lines 164-165: an ADMIXTURE plot with both modern and ancient samples might be good to visualise fine-scale heterogeneity in the ancient samples and their relationships with modern individuals.

We have performed the ADMIXTURE analysis and provided the results in the new SI Figure S2. The ADMIXTURE results are qualitatively consistent with the patterns found in PCA.

Comment 7: Page 5, lines 178-181: can this admixture event be dated using methods like DATES presented in Narasimhan et al, 2019 (<https://zenodo.org/record/3263997#.XRnebJNKj6A>) or ALDER?

We appreciate the reviewer's suggestion. Considering the small sample size of Chokhopani and incomplete coverage, we applied DATES, using Suila and Naxi+Yi as two sources. We obtained an admixture date of 46.1 generations with a leave-one-chromosome-out jackknifing standard error estimate of 10.9 generations, corresponding to 729-2037 years (mean \pm 2 s.e. assuming 30 years per generation) before the time of Chokhopani (around 800 BCE). We report the results in the main text and provide a summary of the DATES results in the new SI Figure S6.

Comment 8: Page 8, lines 299-301 and Table S17: What are the SNPs tested tagging the derived haplotype in EPAS1?

The 19 SNPs tagging the *EPAS1* haplotype are from Huerta-Sanchez et al (2014, Nature). We now provide a new SI Table S18 presenting their rs number, position, and per-SNP coverage information.

Comment 9: Figure 1: is there a way to indicate if samples are from high, low and middle altitudes?

Ancient aMMD individuals reported in this study are all from high altitude regions (2800-4000 masl) as described in the Methods section and SI Text, while the remaining ancient individuals are all from low altitude. Regarding the present-day populations, altitude information of Nepalese and Bhutanese populations can be found in the original studies (Arciero et al., 2018; Jeong et al., 2018; Jeong et al., 2014). Altitude information for the remaining populations are not available, therefore we prefer not to present incomplete altitude information in the figure. Altitude information of the populations, if available, are provided in the revised Table S6.

Comment 10: Figures S2 and S3: I would label the populations with the highest sharing with the ancient samples on the plot or mention it in the caption.

Done.

Comment 11: Figure S4: I would insert a small legend in the figure explaining the different parts of the plot or expand the description in the figure caption.

We extended the Figure S4 legend to better explain the figure.

Reviewer #2 (Remarks to the Author):

Comment 12: In “Ancient genomes from the Himalayas illuminate the genetic history of Tibetans and their Tibeto-Burman speaking neighbors” the authors generate genome wide sequencing and genotyping data on 33 ancient individuals (from approx. 3500 to 2000 years in the past) from high altitude regions on the southern end of the Tibetan plateau. Based on this genetic data they conclude that the population is most closely related to Late Neolithic populations in the region, but also have a Paleolithic Eurasian ancestry. Late Neolithic origin is consistent with prior hypotheses that permanent settlements were only established on the northern edge of the Plateau due to the advent of barley-based agriculture. Paleolithic components of present day Tibetans is also seen in mitochondrial and Y chromosomal haplogroups. They also show by comparing ancient and modern samples, that positive selection has been acting throughout this time period on high altitude adaptive alleles.

The paper is well written and gives a thorough investigation of genetic similarities and source populations for ancient Tibetan and Tibeto-Burman samples. However, the methods and some of their assertions require further explanation/justification.

We respond to the reviewer’s individual comments below.

My comments are as follows:

Comment 13: The ancient data provides a snapshot into the past, but it does not make it a perfect sample of the past since it depends on from where and from what time periods the samples are sourced.

We agree with the reviewer and we take this issue into account while interpreting our results. We believe that our conclusions are robust and insensitive to the representativeness issue because 1) the presence of the Tibetan lineage by the time of Suila (1494-1317 BCE), 2) the deep Eurasian contribution to the ancient and present-day Tibetan lineage populations (tested for both ancient and present-day populations), and 3) the admixture modeling of Tibeto-Burman populations. We acknowledge (on p. 9) that our data cannot completely reject the barley-driven population expansion into the Plateau after 1650 BCE, given that the earliest aMMD individual still post-dates it by ca. 200 years, however, it would require the barley farmers to have migrated more than 1,800 km in just 200 years, a scenario that we find implausible given the nature of the terrain and challenging environment.

Comment 14: 2. In the discussion, the authors state “However, the complex genetic histories of contemporary populations severely hamper the accuracy of inferences based on present-day population.”

However, from my reading, I fail to see how what has been proposed using present-day sample is inaccurate and how the proposed modeling done here provides more accurate inferences? It would be helpful if this could be done in a more statistically principled way.

We rephrased the paragraph to clarify our meaning. The main issues of the published demographic models of Tibetan history inferred from present-day data are: 1) they contradict each other (as

described earlier in the same paragraph) and 2) they simplistically assume present-day Tibetans and Han Chinese to be direct descendants of ancestral Tibeto-Burman and Sinitic lineages, respectively, without further contributions from other lineages. We highlight that our ancient genome-based approach resolves – in part – these issues by 1) providing a direct measure for testing competing hypotheses (e.g., is the differentiation between Tibeto-Burman and Sinitic lineages older than 3,000 years?) and 2) providing better proxies for the actual ancient populations involved in the demographic processes, such as Upper_YR_LN for close ancestors of early barley farmers and Suila/Lubrak for early groups belonging to the Tibetan-related lineage. The following paragraph discusses key implications of our study in detail.

Comment 15: 3. The authors refer to incompatible inferences:

“Tibetans representing a sister clade that split from Han Chinese less than three thousand years ago” and “Tibetans branching off from Paleolithic Siberians (Ust’-Ishim) or even from an unknown archaic hominin.” Could the authors say more as to how these are incompatible? The admixture graphs (from qpGraph) provide evolutionary relationships but not the timing of population splits. So, how does the graph reject a population split of 3000 years ago?

Our qpGraph model shows that the Tibetan lineage cannot be modeled as a sister clade of lowlander groups (e.g. Upper_YR_LN) and requires a contribution from an unknown deep Eurasian lineage. In addition, we show that a clear differentiation between Suila (Tibetan-like) and Upper_YR_LN (lowland) was already present well before 3,000 years ago. These two findings clearly contradict a simple split between Han Chinese and Tibetan < 3,000 years ago.

Comment 16: 4. The authors show us a set of admixture graphs in the supplementary material (Figure S7 and 8), and it would be helpful to let the reader know specifically how these graphs reject previous incompatible results/hypotheses? There is also a set of assumptions and populations that are used to build these graphs that are not justified in the methods section. Also, it was not clear how the graph in the main text was chosen over all the other graphs in the supplemental section, given that these differ in the number of populations used, and model selection is a hard problem when the numbers of parameters differ. The authors claim that some graphs are a better fit than others but the comparisons are not systematically presented.

Our qpGraph analysis focuses on 1) testing if the aMMD and present-day Tibetans/Sherpa can be modeled as a sister clade of a lowland population, and 2) if not, specifying the phylogenetic position of the additional ancestry component of aMMD that is not represented by lowlanders. The backbone graphs include representatives of major East Asian and Native American ancestry components (e.g., Ami, Devil’sGate, Mixe, and Upper_YR_LN), a western Eurasian outgroup (MA-1), and deep-branching populations one at a time (Tianyuan, Hoabinhian, archaic hominins, and Ust’-Ishim; see Fig. S8). The deep branching populations are added into the backbone graph to test their proposed connection with the Tibetan lineage: if the deep Eurasian ancestry of the Tibetan lineage were to come from such a population, we would expect to locate the deep Eurasian gene flow into the Tibetan lineage onto the branch leading to that population. Because none of the tested known deep branches anchors the gene flow into the Tibetan lineage, we conclude that it is from an as yet-to-be-sampled lineage.

The backbone graphs we used were chosen based on our model selection criteria described in the Methods section by: 1) iteratively adding one population at a time, 2) testing all possible 1-way (no admixture) and 2-way admixture models when adding a population, 3) choosing a topology with the least number of f-statistics deviating more than 2 standard errors from the expectation, and 4)

preferring graphs with no zero-length branches. The resulting backbone graphs are compatible with those reported in other studies.

We do not intend to choose graphs across different backbone graphs. Instead, we add each of the aMMD and present-day Tibetans/Sherpa to each backbone graph and choose the best graphs using the same criteria. The resultant graphs consistently support 1) the necessity of a deep branch gene flow into the Tibetan lineage, and 2) a deep Eurasian position of the source (around the split between western and eastern Eurasian lineages). We have updated Figs. 3, S7, S8 to clarify this.

Comment 17: 5. In the results section (lines 223-229), the authors state that “our results reject previously suggested sources of gene flow into the Tibetan lineage, including the deeply branching Eastern Eurasian lineages...in south east Asia (Fig. S8).” There are several graphs in Figure S8, and when I inspect the ones with archaic humans or Ust’ Ishim, I see that there is no contribution from Denisovans into Tibetans, but we know that there is a Denisovan contribution. Is it because this method fails to detect low levels of admixture? I believe what the authors are referring to is results in Lu et al. (2019) - could the authors explain/justify more how these models reject those results?

The contribution from Denisovan is clearly evident in the *EPAS1* region due to the impact of positive natural selection. However, the Denisovan proportion in the rest of the genome, which evolves neutrally or is weakly affected by negative selection, is extremely small: previous studies report ~0.5% at the genome-wide level (Mallick et al., 2016, Nature 538: 201-206). Therefore, we believe that it does not significantly affect the results to ignore a Denisovan admixture in Tibetans.

Comment 18: 6. How about some of the authors’ previous work (e.g. Jeong et al. Admixture facilitates genetic adaptations to high altitude in Tibet. Nat. Commun. 5, 1–7 (2014))? How do their previous results stand in light of the new ancient DNA and new analyses presented here?

This work is in line with the findings reported in our previous analyses of the 8 aMMD samples, confirming these individuals show strong affinities to present-day Tibetan populations compared to lowland Tibeto-Burman speakers. Also, we replicate admixture signals and the east-west admixture cline of Tibetan populations we previously reported (Jeong et al., 2014 Nat Commun; Jeong et al., 2017 PLoS One). In addition, we utilize a greater time span and sample sizes to highlight heterogeneity between aMMD individuals and differential relationship between aMMD groups to present-day populations. We have updated the Discussion section (p. 8) to clarify links between the current study and our previous ones.

Other comments:

Comment 19: 1) It would be good to have a clearer explanation/discussion of how the qpGraph modeling source populations were chosen for present day Tibetan admixture analysis (section starting Line 192 and Fig 3) - it appears to be related to pairwise genetic affinity estimates but it’s not clear how such a complex model arose.

Please see our responses to the above comment 20.

Comment 20: 2) Regarding the unresolved deep Eurasian lineage that shows admixture in the aMMD groups - is there evidence for this admixed lineage by applying other tools to detect such features?

We do not provide additional evidence independent of qpGraph and f-statistics. Our previous studies (Jeong et al. 2014 Nat Commun) reported a split time between Tibetans and lowlanders that is much older than the post-1650 BCE date expected for the barley hypothesis, based on the pairwise sequentially Markovian Coalescent (PSMC) analysis. The Tibetan-specific Denisovan-related admixture, which brought the *EPAS1* haplotype, as well as the presence of Y chromosome haplogroup D-M174 in Tibetans are frequently considered as supporting evidence for the presence of the pre-Neolithic genetic substratum. We discuss these in the Introduction.

Comment 21: 4) For clinal analysis of Figure 4, is Tsum being used also as a source (as noted in the text). If so that should be noted in the caption. It might also be more visually coherent/appealing to present this in the view of Figure 1 (with contours for the cline and source populations indicated)

We have updated figure 4 and its legend to clarify that we used Tsum (a Nepalese Tibetan group) and Upper_YR_LN as two sources. We provide figure 5 as a visual summary of admixture modeling results on the map; therefore, we prefer to keep figure 1 as it is.

Comment 22: 5) how is f_3 different than the population branch statistic?

Our outgroup- f_3 and PBS perform in a similar way, if we set up PBS with present-day Tibetan, aMMD, and Han Chinese as the target, comparison group, and outgroup, respectively. We chose outgroup- f_3 because allele frequency estimates of aMMD are noisy due to incomplete coverage and relatively small sample size: PBS requires calculating F_{ST} for both Tibetan-aMMD and Han-aMMD pairs. Because both of these pairs will be noisy, we prefer a simple statistic of outgroup- f_3 that is less noisy.

Comment 23: 6) Line 284: “Interestingly, the derived allele frequency in the ancient samples overall is lower than in present-day Tibetans (75%).” The frequency of Tibetans varies geographically, and I believe is correlated with altitude. What is the altitude of the samples being used for the present-day Tibetan populations?

The present-day Nepalese Tibetans are UpperMustang, Tsum, and Nubri, with average altitudes of 3,421, 3,436, and 3,746 m, respectively. Previous studies report similarly high *EPAS1* haplotype frequency among Tibetans in the Plateau: e.g. 77-81% (3810-4200 m; Peng et al., 2011 Mol Biol Evol 28: 1075-1081) and 64.5% (3650 m; Hackinger et al., 2016 Hum Genet 135: 393-402). We believe that the present-day Nepalese Tibetans are the most appropriate targets of comparison because they are geographically closest to aMMD among published Tibetan samples. We further cited Jeong et al. 2018 for clarification.

Comment 24: Typos: Line 233 - Two-routes *of* dispersal....

Corrected.

Reviewers' Comments:

Reviewer #1:

Remarks to the Author:

The authors have addressed my comments and clarified the details of some the analyses performed. I think this revised version has improved compared to the original submission.

Reviewer #4:

Remarks to the Author:

This manuscript presents genetic history analyses of 33 ancient DNA samples of Tibetans from the South side of the Himalayans. A major question left from previous studies is who are the present-day Tibetans – whether they are descendants of Neolithic farmers, or if (and to what extent) they are related to the Paleolithic Hunter-Gatherers on the Tibetan Plateau. Although many previous studies tried to answer this question by estimating genetic affinities and the time of settlement using modern genomes, the large intervals of genetic estimation of very old events left open several plausible hypotheses that only aDNA can provide insight into. For the most time, there were few aDNA samples from the high-altitude regions of TP. Therefore, in this area, any new sample is potentially game changer, and this study increased the available ancient sample size by several folds. Although these samples were from a relatively recent time range (<3500 years ago), they still filled a void in the study of Tibetan population history. And I personally very appreciate the discussion at the end regarding the hypothesis of agriculture-facilitated Neolithic population expansion, which is particularly interesting as the advent of Barley-based agriculture was around the same age as the oldest sample from this study.

I see that the authors appropriately and sufficiently addressed the previous two reviewer comments, and I don't have major analysis to suggest. I only have a minor comment:

For the introgressed EPAS1 allele, is it possible to estimate the start time of positive selection on it using allele frequency time-series data, now that the authors have its allele frequency from ~3000 years ago? Further, how does the length of the introgressed haplotype in these ancient individuals compare to the modern Tibetans?

Reviewer #5:

Remarks to the Author:

This paper provides one of the most comprehensive aDNA studies of proto-Tibetan populations to date and its findings confirming the genetic ancestry of modern day Tibetans with Neolithic-early Historic groups in Nepal. Genome wide selection scan across a good sample of individuals in the study also attest to selection at EPAS1 loci in ancient populations, thus providing a crucial piece of evidence showing the rise in this allele frequency among highland populations. In this respect, the study contributes significantly to understanding a poorly understood population history.

One of the claims concerning the putative ancestral population from the Late Neolithic in northwest China needs to be clarified. One of the main hypothesis re: origins attributes the arrival of groups in the plateau with barley farmers from northwest China. The timing of this expansion remains under debate. The authors think the expansion happened earlier than the 'consensus' hypothesis of 2000 BC (see lines 370-1) because the Neolithic comparison group - Qijia - in NW China did not share the EGLN1 gene. The absence of EGLN1 signature in this northwest population is however not surprising. These were lowland valley agriculturalists, possibly part time pastoralists. After 3600 BP, the Qijia culture was followed by the Kayue culture (3600-2600 BP). This late Neolithic group, which overlaps in time with the two early and three late MMD individuals from this study, moved into higher elevations

from the lower gradients of its Neolithic predecessors - the Qijia and Majiayao - and practiced classic transhumance along with shifting cultivation of wheat and barley.

To substantiate a claim for an earlier migration, the authors would either need aDNA materials from early Neolithic groups in NW China or demonstrate that the MMD and Kayue peoples are not related. This leads one to wonder about the archaeological context - what kinds of tomb structures and material artifacts were these MMD individuals buried with? Is their material repertoire comparable with the Qijia or the Kayue? I'm not convinced that the evidence presented refutes the consensus hypothesis.

Point-by-Point Response to Editor’s and Reviewers’ comments for NCOMMS-21-01104-A
“Ancient genomes from the Himalayas illuminate the genetic history of Tibetans
and their Tibeto-Burman speaking neighbors”

We received five comments from three reviewers. We present our response to each comment below.

Reviewer #1 (Remarks to the Author):

Comment 1: The authors have addressed my comments and clarified the details of some the analyses performed. I think this revised version has improved compared to the original submission.

We appreciate the reviewer’s positive assessment of our study.

Reviewer #4 (Remarks to the Author):

Comment 2: This manuscript presents genetic history analyses of 33 ancient DNA samples of Tibetans from the South side of the Himalayans. A major question left from previous studies is who are the present-day Tibetans – whether they are descendants of Neolithic farmers, or if (and to what extent) they are related to the Paleolithic Hunter-Gatherers on the Tibetan Plateau. Although many previous studies tried to answer this question by estimating genetic affinities and the time of settlement using modern genomes, the large intervals of genetic estimation of very old events left open several plausible hypotheses that only aDNA can provide insight into. For the most time, there were few aDNA samples from the high-altitude regions of TP. Therefore, in this area, any new sample is potentially game changer, and this study increased the available ancient sample size by several folds. Although these samples were from a relatively recent time range (<3500 years ago), they still filled a void in the study of Tibetan population history. And I personally very appreciate the discussion at the end regarding the hypothesis of agriculture-facilitated Neolithic population expansion, which is particularly interesting as the advent of Barley-based agriculture was around the same age as the oldest sample from this study.

I see that the authors appropriately and sufficiently addressed the previous two reviewer comments, and I don’t have major analysis to suggest. I only have a minor comment:

We appreciate the reviewer’s positive assessment of our study.

Comment 3: For the introgressed *EPASI* allele, is it possible to estimate the start time of positive selection on it using allele frequency time-series data, now that the authors have its allele frequency from ~3000 years ago? Further, how does the length of the introgressed haplotype in these ancient individuals compare to the modern Tibetans?

After further analysis, we find that selection still acted on *EPASI* alleles in the recent past because each of the aMMD groups consistently has a lower derived allele frequency compared to present-day Tibetans,

resulting in a significant difference between aMMD as a group and present-day Tibetans. However, each aMMD group on its own consists of a limited number of individuals, which leads to noisy estimates of allele frequencies at specific time points (Fig. S13). We thus refrain from estimating the onset of selection based on time series data.

Regarding the length of the introgressed haplotype, we investigated if there were novel variants found around the core Denisovan segment among the aMMD individuals, especially those matching the Denisovan allele. However, we did not find such novel variants, concluding that the *EPASI* haplotype of the aMMD individuals had the same length of the introgressed segment as that of present-day Tibetans.

Reviewer #5 (Remarks to the Author):

Comment 4: This paper provides one of the most comprehensive aDNA studies of proto-Tibetan populations to date and its findings confirming the genetic ancestry of modern day Tibetans with Neolithic-early Historic groups in Nepal. Genome wide selection scan across a good sample of individuals in the study also attest to selection at *EPASI* loci in ancient populations, thus providing a crucial piece of evidence showing the rise in this allele frequency among highland populations. In this respect, the study contributes significantly to understanding a poorly understood population history.

We appreciate the reviewer's positive assessment of our study.

Comment 5: One of the claims concerning the putative ancestral population from the Late Neolithic in northwest China needs to be clarified. One of the main hypothesis re: origins attributes the arrival of groups in the plateau with barley farmers from northwest China. The timing of this expansion remains under debate. The authors think the expansion happened earlier than the 'consensus' hypothesis of 2000 BC (see lines 370-1) because the Neolithic comparison group - Qijia - in NW China did not share the *EGLN1* gene. The absence of *EGLN1* signature in this northwest population is however not surprising. These were lowland valley agriculturalists, possibly part time pastoralists. After 3600 BP, the Qijia culture was followed by the Kayue culture (3600-2600 BP). This late Neolithic group, which overlaps in time with the two early and three late MMD individuals from this study, moved into higher elevations from the lower gradients of its Neolithic predecessors - the Qijia and Majiayao - and practiced classic transhumance along with shifting cultivation of wheat and barley.

To substantiate a claim for an earlier migration, the authors would either need aDNA materials from early Neolithic groups in NW China or demonstrate that the MMD and Kayue peoples are not related. This leads one to wonder about the archaeological context - what kinds of tomb structures and material artifacts were these MMD individuals buried with? Is their material repertoire comparable with the Qijia or the Kayue? I'm not convinced that the evidence presented refutes the consensus hypothesis.

We agree with the reviewer that the ancient genomes associated with the Kayue culture (who remain genetically unsampled) would provide a more robust test for the relationship between the early barley farmers and the early aMMD. In this study, we explored this topic using the ancient genomes from the Qijia culture, an immediate predecessor of the Kayue culture, as a plausible proxy for the genetic profile

of the Kayue culture people. While we cannot decisively exclude a scenario in which the Kayue people had a genetic profile that was distinct from the Qijia people but matching the early aMMD (Suila/Lubrak) (i.e. a scenario in which the Kayue people did not genetically descend from Qijia), we believe that this is a highly unlikely scenario given the following genetic and archaeological data, which are summarized below:

- 1) As described in the text (especially in Text S1), the early aMMD sites show mortuary practices distinct from those of both the Qijia and Kayue lowlanders. Suila is a cave tomb, while Lubrak consists of two slab-cist burial chambers that were likely components of a shaft tomb. Rhirhi mortuary patterns reflect a cave burial tradition with close structural similarities to those found in far western Tibet, and it has artifact assemblages with strong affinities with South Asia (see Text S1). The ceramics from Lubrak are clearly related to those found in Ladakh and the western Himalayas, while those from Rhirhi consist of locally produced ceramics. None of these mortuary patterns and ceramic assemblages have any similarity to those found in Qijia or Kayue contexts (see Dong et al. 2013 *J Archaeol Sci* 40: 2538-2546 for a brief description of these assemblages).
- 2) Likewise, recent archaeological findings in the Tibetan Plateau also support an early archaeological tradition in the Plateau distinct from the lowland Qijia/Kayue cultures. In the main text, we discussed archaeological evidence from the Zongri site and related sites, which suggests the presence of a plateau-based foraging group that traded for millets with lowlanders (lines 369-371). Zongri sites range in date from ca. 2600-2000 BCE (Ren et al 2020 *Antiquity* 94: 637-652). Zongri mortuary patterns are distinct from those of the lowlanders with whom they traded, and there are clear differences in the artifact assemblages and mortuary patterns between the two cultures. Also, studies on the Karou site in eastern Tibet (ca. 5000-3000 BP) and the Qugong site near Lhasa (ca. 3800-3000 BP) show assemblage composition and ceramic motifs wholly distinct from those at Qijia and Kayue (d'Alpoim Guedes and Aldenderfer 2020 *J Archaeol Res* 28: 339-392). We now more clearly highlight these points in the Discussion section of the main text.
- 3) Previous genetic studies dated the onset of selection on *EGLNI* at ~8,000 BP (lines 371-372), which is much older than the time of the Qijia culture (4,000 BP). If the early aMMD split from the Qijia people, the common ancestor of the Qijia and aMMD must have experienced selection on *EGLNI* for over four millennia. Therefore, lack of the *EGLNI* variants in these lowland populations argues against the scenario that aMMD recently split from the Qijia people.
- 4) As we discussed in the main text (see Discussion), the earliest dates of the aMMD (Suila, 1494-1317 BCE) falls close in time to the earliest evidence for extensive barley farming by the lowlanders in the northeast margin (ca. 1650 BCE). Considering ~1,600 km of the linear distance from the nearest Qijia or Kayue sites to Upper Mustang, the reviewer's scenario requires an extremely rapid spread/migration of a cultural complex from northwestern China to Upper Mustang within the ~200-year period regardless whether the source was Qijia or Kayue. While such a rapid migration is certainly within the realm of possibility, it is highly improbable. We acknowledge both scenarios in the text, but note the differences in plausibility based on the radiocarbon dating.